# Decomposing Scientific Paper Queries with Draft-and-Follow Policy Optimization to Narrow Knowing-Doing Gap

## Abstract

The rapid growth in the volume of scientific papers presents a significant challenge for researchers to keep up with the latest advances in their field by relying solely on manual reading. Given recent advances in Large Language Models (LLMs), there is a growing trend of employing autonomous agents to extract key information from scientific papers. Although promising, existing approaches generally rely on either meticulously engineered prompts or a standard SFT-RL pipeline, methodologies that are often prone to inducing excessive and ineffective exploration. Inspired by cognitive science, we introduce **PaperCompass**, a novel framework designed to address these limitations. Specifically, PaperCompass first generates a draft outlining the sequence of planned execution steps and subsequently engages in fine-grained reasoning to determine parameters for the corresponding function calls. Furthermore, to support this process, we develop a bespoke RL method named **D**raft-**F**ollow **P**olicy **O**ptimization, which concurrently optimizes both the draft plan and the final solution. **DFPO** can be viewed as a streamlined implementation of Hierarchical RL, designed to bridge the 'knowing-doing' gap observed in LLMs. We provide a theoretical analysis of DFPO, demonstrating its desirable properties and thereby ensuring a reliable optimization process. Experiments on paper-based question-answering (Paper-QA) benchmarks demonstrate that PaperCompass's superior efficiency over existing baselines without compromising performance, achieving results comparable to those of much larger models.

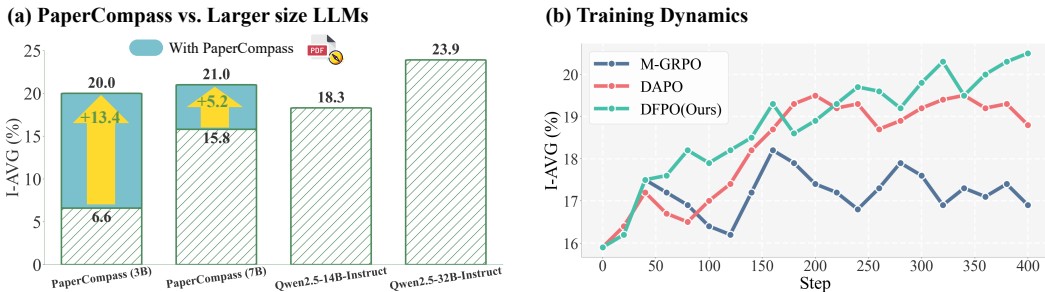

Figure 1: The performance of PaperCompass on AirQA-Real (Cao et al., 2025), and the training dynamics of DFPO compared with other RL training algorithms initialized from DTFT (Draft-and-Follow Fine-Tuning), 3B size. **(a):** Our PaperCompass achieves performance comparable to that of much larger 32B-parameter baseline models. **(b):** DFPO has demonstrated even more powerful efficiency.

## 1 Introduction

Agents built upon the powerful capabilities of Large Language Models (LLMs) offer researchers significant benefits, such as automating literature retrieval (He et al., 2025; Chen, 2025; Shi et al., 2025), scientific discovery (Lu et al., 2024; Liao et al., 2024; Zhao et al., 2024; Wysocki et al.,

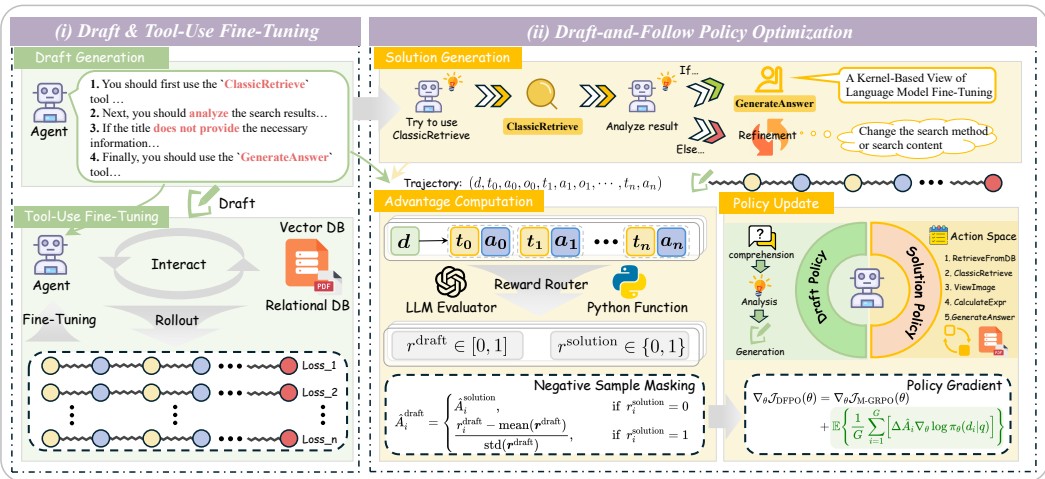

Figure 2: **PaperCompass**. (i) We use Qwen2.5-32B-Instruct to generate complete trajectories containing draft and solution on synthetic data to fine-tune the agent. This step enables the agent to understand the basic task logic and the tool calling format. (ii) DFPO facilitates the hierarchical optimization of both the initial draft and the subsequent solution, uniquely achieving this bi-level refinement by maximizing a single objective function.

2024), and even automatic report generation (Schmidgall et al., 2025; Ferrag et al., 2025; Team et al., 2025). Consequently, researchers are increasingly turning to LLMs for assistance. However, a more pressing challenge for many researchers is the efficient extraction of key information from state-of-the-art papers. The rapid growth in the volume of academic literature makes it increasingly difficult for researchers to keep abreast of the latest developments in their fields solely through manual reading.

A direct approach is to employ general-purpose LLMs (Achiam et al., 2023; Anthropic, 2024; Guo et al., 2025; Comanici et al., 2025; Yang et al., 2025) within the prevailing paradigm for question answering over large corpora, namely Retrieval-Augmented Generation (RAG) (Lewis et al., 2020; Guu et al., 2020; Biswal et al., 2024; Cao et al., 2025). It's well-established that applying advanced RAG methods, such as Agentic RAG, to large size models yields strong performance on certain QA benchmarks. However, the efficacy of these methods often diminishes when applied to smaller-scale models (e.g., those in the 3B class). A prominent failure mode in such cases is the agent entering an unproductive exploratory loop, typically characterized by the repeated execution of an ineffective retrieval strategy (Deng et al., 2025). This failure mode is consistent with recent findings (Paglieri et al., 2024; Ruoss et al., 2024; Schmied et al., 2025) on the 'knowing-doing' gap in language models, which we posit as the underlying cause. Specifically, 'knowing-doing' gap refers to the phenomenon where, even for correctly computed rationales, the model often selects the greedy action over the optimal one. This discrepancy highlights the shortcomings of the LLM when it comes to 'doing' even when 'knowing' the decision process. In practice, the 'knowing–doing gap' manifests in Paper-QA tasks in very concrete ways. For example, the model may correctly identify that the answer is likely located in a different section, yet repeatedly issues the same retrieval action, or continues searching even when sufficient evidence has already been gathered. This mismatch between "knowing what to do" and "actually doing it" results in unproductive action loops, especially for smaller models. This directly motivates a framework that separates reasoning ('knowing') from execution ('doing') and optimizes them with distinct credit signals. Furthermore, the inherent challenges of the Paper-QA task, such as *Cross-Section Dependencies*, *Dense and Specialized Content*, and *High Factual Precision*, further compound the decision-making difficulties for smaller-scale models. Therefore, a central research question arises: *How can we design training frameworks that enable agents, especially based on smaller-scale models, to both learn essential meta-capabilities and reliably execute them throughout long-horizon decision-making tasks?*

In light of these challenges, we propose **PaperCompass**, a novel multi-turn RL framework for training agents for scientific paper querying, with its overall architecture shown in Figure 2. Inspired by cognitive science (Ho et al., 2022) and recent work on LLM Reasoning (Zhang et al., 2025a; Kang et al., 2025), PaperCompass explicitly bridges the 'knowing-doing' gap by requiring the agent to first construct a high-level plan—termed a **draft**—before executing a sequence of fine-grained actions based on the ReAct framework (Yao et al., 2023). To train this hierarchical process, we introduce **D**raft-and-**F**ollow **P**olicy **O**ptimization (DFPO), a bespoke RL algorithm that functions as a streamlined implementation of Hierarchical RL. DFPO concurrently optimizes both the quality of the draft ('knowing') and the fidelity of the subsequent solution ('doing') by maximizing a single objective function. Our theoretical analysis shows that DFPO's policy gradient can be decomposed into the M-GRPO(Shao et al., 2024; Wei et al., 2025) gradient plus an implicit bias term induced by DFPO's draft–solution structure, and under certain conditions, is guaranteed to assign an advantage bonus to optimal drafts. Experimentally, we develop a novel efficiency-aware metric, I-Avg, on which our 3B model achieves performance comparable to a 32B model, as partly shown in Figure 1. Furthermore, DFPO significantly outperforms other RL methods on standard metrics.

## 2 PRELIMINARIES AND BACKGROUND

**Problem Formulation.** We consider adopting an ReAct-Paradigm LLM (Yao et al., 2023) as an autonomous agent to answer different questions for scientific papers. Upon receiving a question, the agent performs several iterations of Thought-Action-Observation. In each iteration, the agent first analyzes the current context to determine and execute an action, formulated as a specific tool call. Following this, the agent receives a new observation from the environment, reflecting the outcome of the tool's interaction. Consistent with NeuSym-RAG (Cao et al., 2025), we utilize the 5 parameterized actions with arguments that agents can take during interaction, which are detailed in Appendix A.1. The iteration terminates when the LLM selects `GenerateAnswer` as the action. A complete trajectory with $N$ iterations can be defined as follow:

$$\mathcal{T}_N = (t_0, a_0, o_0, \cdots, t_{N-1}, a_{N-1}, o_{N-1}, t_N, a_N),$$

where $(t, a, o)$ represents a tuple of Thought-Action-Observation.

**Multi-turn GRPO.** GRPO (Shao et al., 2024) has been widely applied in the field of LLM Reasoning due to its outstanding performance and ingenious design. Unlike single-turn reasoning, trajectories in Agentic-RL incorporate environmental observations. However, these observations are merely intermediate steps and not the primary objective of optimization. WebAgent-R1 (Wei et al., 2025) formally proposed the Multi-turn GRPO. For each question $q$, the agent first sample a group of trajectories $\{\tau_1, \tau_2, \cdots, \tau_G\}$ and then optimize the policy model $\pi_\theta$ by maximizing the following objective function:

$$\mathcal{J}_{\text{M-GRPO}}(\theta) = \mathbb{E}_{q \sim P(q), \{y_i\}_{i=1}^{G} \sim \pi_{\text{old}}(\cdot|q)} \left\{ \frac{1}{G} \sum_{i=1}^{G} \left( \frac{1}{|y_i|} \sum_{t=1}^{|y_i|} \left[ \hat{A}_{i,t} - \beta \, \mathbb{D}_{\text{KL}}(\theta) \right] \right) \right\},$$

where $y_i = (t_{i,j}, a_{i,j})_{j=0}^{N}$ is the generated outputs of LLM and $\hat{A}_{i,t}$ is the advantage for the $t$-th token of $i$-th trajectory:

$$\hat{A}_{i,t} = \min \left\{ \frac{\pi_\theta(y_{i,t}|q, y_{i,<t})}{\pi_{\text{old}}(y_{i,t}|q, y_{i,<t})} \hat{A}_i, \text{clip}\left( \frac{\pi_\theta(y_{i,t}|q, y_{i,<t})}{\pi_{\text{old}}(y_{i,t}|q, y_{i,<t})}, 1 - \epsilon, 1 + \epsilon \right) \hat{A}_i \right\},$$

$\hat{A}_i = \frac{r_i - \text{mean}(\boldsymbol{r})}{\text{std}(\boldsymbol{r})}$ is the group relative advantage, computed using a group of rewards $\boldsymbol{r} = \{r_1, \cdots, r_G\}$ produced by rule-based reward functions.

## 3 PAPERCOMPASS

### 3.1 DRAFT-AND-FOLLOW: AN ANALOGOUS FRAMEWORK TO OPTION

Inspired by human planning in cognitive science (CAMBRIDGE; Eysenck & Keane, 2020; Ho et al., 2022) and recent LLM Reasoning research (Zhang et al., 2025a; Kang et al., 2025), we introduce

the Draft-and-Follow framework. It aims to narrow the 'knowing-doing' gap (Paglieri et al., 2024; Ruoss et al., 2024; Schmidgall et al., 2025), where LLMs fail to execute optimal plans and instead select greedy actions. Our framework imposes a hierarchical decision-making process: A high-level draft leverages the LLM's instruction-following capabilities to guide the low-level follow execution, thereby preventing deviations into greedy action sequences.

Our framework is adapted from the option framework in Hierarchical RL (HRL) (Sutton et al., 1999b; Bacon et al., 2017). In HRL, an option $\omega$ is a temporally abstract action defined by a tuple $(\mathcal{I}_\omega, \pi_\omega, \beta_\omega)$, representing the initiation set, the intra-option policy, and the termination function, respectively. In our **Draft-and-Follow** framework, the draft serves as an analogue to the intra-option policy $\pi_\omega$. As illustrated in Figure 3, a high-level draft (e.g., *retrieval* followed by *analysis*) corresponds to a sequence of low-level tool calls and reasoning steps. Crucially, the initiation set $\mathcal{I}_\omega$ and termination function $\beta_\omega$ are not explicitly defined but are handled dynamically by the LLM's reasoning, which is similar to Feng et al. (2024). An option is initiated when the agent determines more information is required and is terminated once sufficient information is gathered to form a solution.

Notably, a key distinction of our proposed Draft-and-Follow framework is that it integrates two functional roles within a single agent, in contrast to the classic option framework which often requires distinct sub-agents. This is possible due to the inherent versatility of LLMs compared to traditional RL agents, which typically execute a singular policy. Consequently, our single LLM-based agent can dynamically switch between roles: First acting as a high-level planner to generate the draft, and subsequently as a low-level executor to implement that plan and produce the final solution.

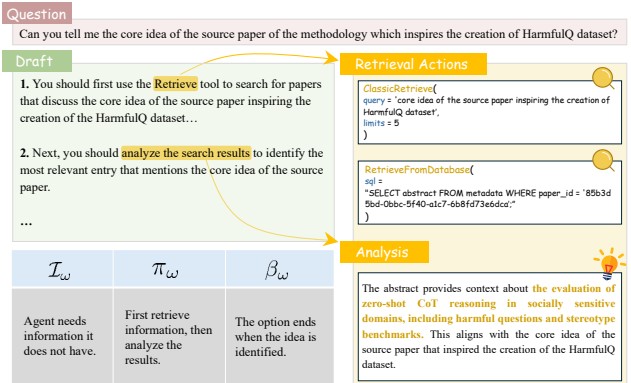

Figure 3: The conceptual relationship between the option framework and Draft-and-Follow framework.

## 3.2 DRAFT & TOOL-USE FINE-TUNING

The agent must acquire two foundational capabilities. First, it must be capable of generating syntactically correct tool calls. This includes accurately selecting function names and formulating their corresponding parameters from the action space. Second, the agent must learn to generate a coherent draft and then faithfully execute this plan in the subsequent solution. Due to the prohibitive cost of manually annotating expert trajectories, our approach relies on the generation of a synthetic dataset. we construct synthetic expert trajectories following a three-stage pipeline.

❶Explorer generates high-quality question–answer pairs from 10,000 arXiv AI papers, using CoT prompting and curated templates to ensure diversity and fidelity. ❷Actor interacts with the paper database via a ReAct-style LLM agent to produce full action trajectories, including tool calls, intermediate thoughts, and error states. ❸Tracker converts these trajectories into structured training examples by selecting appropriate evaluation functions, formatting answers, and packaging long-horizon interactions using a sliding window. Finally, an expert LLM summarizes each trajectory into an abstract, high-level draft plan to establish a one-to-one mapping between drafts and expert solutions. Further details are provided in Appendix A.2.

## 3.3 DRAFT-AND-FOLLOW POLICY OPTIMIZATION

As our previous analysis has shown, the draft is a central component of our framework, providing high-level guidance for the agent's subsequent tool calls. This central role motivates the need for a novel optimization objective. An effective objective must not only optimize the draft and the solution concurrently but also place a distinct emphasis on improving the quality of the draft itself. To meet

these requirements, we introduce a new optimization objective. We term the process of optimizing this objective Draft-and-Follow Policy Optimization (DFPO). Formally, the goal is to maximize the following objective function:

$$\mathcal{J}_{\text{DFPO}}(\theta) = \mathbb{E}_{\substack{q \sim P(q) \\ \{d_i, y_i\}_{i=1}^G \sim (\pi_\theta(d|q), \pi_\theta(y|q,d))}} \left\{ \frac{1}{G} \sum_{i=1}^{G} \left[ \frac{1}{|d_i| + |y_i|} \left( \sum_{t=1}^{|d_i|} \hat{A}_{i,t}^{\text{draft}} + \sum_{t=|d_i|+1}^{|d_i|+|y_i|} \hat{A}_{i,t}^{\text{solution}} \right) \right] \right\},$$
(1)

where $d$ is the draft while $y$ is the solution, which is also corresponds to the generated outputs of LLM mentioned in Section 2. We have abandoned the KL divergence constraint and generated trajectories entirely using the current policy (fully on-policy). These two points have been proven by recent studies to be able to effectively improve the performance of the agent (Lanchantin et al., 2025; Yu et al., 2025). For a given QA instance, we let $o = d \circ y$ denotes the complete output sequence from the agent. By definition, the draft is a prefix of the full response $o$. This structural property allows for the formulation of the policy gradient with respect to the draft as follows:

**Proposition 1** (Gradient Decomposition). *In a fully on-policy setting[1] without a KL divergence constraint, the DFPO policy gradient can be written as the sum of the M-GRPO policy gradient and an additional term arising from the draft–solution structure of the DFPO objective,*

$$\nabla_\theta \mathcal{J}_{\text{DFPO}}(\theta) = \nabla_\theta \mathcal{J}_{\text{M-GRPO}}(\theta) + \mathbb{E}_{q \sim P(q), \{d_i\}_{i=1}^G \sim \pi_\theta(d|q)} \left\{ \frac{1}{G} \sum_{i=1}^{G} \left[ \Delta \hat{A}_i \nabla_\theta \log \pi_\theta(d_i|q) \right] \right\}, \quad (2)$$

*where $\Delta \hat{A}_i = c \cdot (\hat{A}_i^{draft} - \hat{A}_i^{solution})$ is a productive of a scaling constant $c$ and a relative advantage term.*

> **Remark on Proposition 1**
>
> Proposition 1 (see proof in Appendix B.1) provides an interpretive decomposition of the DFPO gradient: It separates the standard M-GRPO update from an implicit bias term induced by DFPO's design—namely, (i) jointly normalizing the draft and solution, and (ii) defining the draft as a prefix of the full response.

This leads to a key design question: *how to formulate distinct reward functions for the draft and the solution such that the DFPO algorithm can improve draft quality while simultaneously optimizing for overall task success?* Specifically, we formulate separate reward signals for both the draft and the solution, and from these, we derive their respective advantage functions:

$$r_i^{\text{draft}} = \rho(d_i) \cdot \mathbb{I}(\texttt{Extract}(y_i) = \text{Answer}) \quad \Rightarrow \quad \hat{A}_{i,t}^{\text{draft}} = \frac{r_i^{\text{draft}} - \text{mean}(\boldsymbol{r}^{\text{draft}})}{\text{std}(\boldsymbol{r}^{\text{draft}})}$$

$$r_i^{\text{solution}} = \mathbb{I}(\texttt{Extract}(y_i) = \text{Answer}) \quad \Rightarrow \quad \hat{A}_{i,t}^{\text{solution}} = \frac{r_i^{\text{solution}} - \text{mean}(\boldsymbol{r}^{\text{solution}})}{\text{std}(\boldsymbol{r}^{\text{solution}})}$$

Here, $\rho(d_i) \in [0, 1]$ is a non-negative dense metric to evaluate the draft. We use $\texttt{Extract}$ to denote the extraction function to derive predicted answer. In our practical implementation, to account for diverse question types, we introduce a reward router (detailed in the Appendix C) that provides a more accurate evaluation of the agent's output quality, thus yielding higher-fidelity gradient signals. Based on the preceding definitions, we now state the following theorem regarding relative advantage. This theorem establishes key theoretical properties of our DFPO algorithm:

**Theorem 1** (Relative Advantage). *If $r^{solution}$ is a binary value, and $r^{draft}$ is a product of $r^{solution}$ and a non-negative dense value, there are:*

- *Let $i$ be an index such that $r_i^{solution} = 0$. The advantage $\hat{A}_i^{solution}$ and $\hat{A}_i^{draft}$ satisfies the inequality:*

$$\hat{A}_i^{solution} \leq \hat{A}_i^{draft}.$$

---

[1]An analysis of DFPO's performance using off-policy rollouts is presented in Appendix B.1

- *Let $i$ be an index such that $r_i^{solution} = 1$. Given that $r_i^{draft} \leq \bar{r}^{draft}$, the advantage satisfies the inequality:*

$$\hat{A}_i^{draft} \leq \hat{A}_i^{solution}.$$

- *Let $i_{\max}$ be an index such that $r_{i_{\max}}^{draft} = \max_{1 \leq i \leq n}\{r_i^{draft}\}$. Given that $r_{i_{\max}}^{solution} = 1$, the advantage satisfies the inequality:*

$$\hat{A}_{i_{\max}}^{solution} \leq \hat{A}_{i_{\max}}^{draft}.$$

> **Remark on Theorem 1**
>
> Theorem 1 (see proof in Appendix B.2) reveals that, when a solution fails, this mechanism provides a non-negative update to reinforce the draft. Conversely, for successful solutions that follow below-average drafts, a non-positive update suppresses these ineffective plans. Finally, when a high-quality draft leads to a successful outcome, a non-negative update reinforces these effective patterns, consolidating the model's successful policies.

In our implementation of PaperCompass, we utilize trajectory-level entropy (Agarwal et al., 2025) as the metric for draft quality. In this way, the non-negative dense metric is:

$$\rho(d_i) = \frac{1}{|d_i|} \sum_{t=1}^{|d_i|} \log \pi_\theta(d_{i,t}|q, d_{i,<t}). \tag{3}$$

A potential concern is that: In Theorem 1, if an incorrect solution stems from a low-quality draft, the standard DFPO objective might inadvertently reinforce this suboptimal plan. To mitigate this issue, we addtionally introduce a technique we term **negative sample masking**, which modifies the draft advantage calculation as follows:

$$\hat{A}_i^{\text{draft}} = \begin{cases} \hat{A}_i^{\text{solution}}, & \text{if } r_i^{\text{solution}} = 0 \\ \dfrac{r_i^{\text{draft}} - \text{mean}(\boldsymbol{r}^{\text{draft}})}{\text{std}(\boldsymbol{r}^{\text{draft}})}, & \text{if } r_i^{\text{solution}} = 1 \end{cases} \tag{4}$$

Negative sample masking also provides further justification for employing trajectory-level entropy as a quality metric (further discussion is detailed in Appendix D). This aligns with related research suggesting that the output logits of a sufficiently pre-trained LLM can implicitly represent a Q-function (Wulfmeier et al., 2024; Wang et al., 2025a; Li et al., 2025b).

## 4 EXPERIMENTS

Our evaluation of PaperCompass on two challenging benchmarks is guided by the following research questions:

- **RQ1:** To what extent does PaperCompass improve interaction efficiency without compromising task performance? (Section 4.2, Section 4.3, Appendix A.4.1, Appendix A.4.2)
- **RQ2:** What is the magnitude of the efficiency improvement that DFPO provides over RL baselines like M-GRPO and DAPO? What are the individual contributions of DFPO's key components? (Section 4.4, Appendix A.4.3)
- **RQ3:** How does the 'Draft' mechanism contribute to the overall efficacy of the PaperCompass framework? (Section 4.5)
- **RQ4:** Is our DTFT a critical component, or does a standard SFT approach provide a sufficient foundation for the subsequent RL training stage? (Appendix A.4.4)

### 4.1 EXPERIMENT SETUP

**Benchmarks.** To comprehensively evaluate the effectiveness of our PaperCompass, we select two prominent benchmarks for scientific Paper-QA: AirQA-Real (Cao et al., 2025), and SciDQA (Singh et al., 2024). Detailed descriptions of these benchmarks are provided in Appendix E.

Table 1: Overall performance on two prominent and challenging Paper-QA benchmarks are presented. The top two outcomes in finetuning methods are **bolded** and underlined.

| Method | AirQA-Real | | | | | | | SciDQA | | | | |
|---|---|---|---|---|---|---|---|---|---|---|---|---|
| | text | table | image | form. | meta. | **Avg.** | **I-Avg.** | table | image | form. | **Avg.** | **I-Avg.** |
| *Prompting Methods with Classic RAG* | | | | | | | | | | | | |
| **Qwen2.5-3B-Instruct** | 6.8 | 0.0 | 0.0 | 2.8 | 4.5 | 6.1 | / | 35.6 | 37.0 | 33.9 | 35.2 | / |
| **Qwen2.5-7B-Instruct** | 8.1 | 0.0 | 2.5 | 2.8 | 4.5 | 7.4 | / | 45.1 | 44.9 | 45.4 | 45.1 | / |
| **Qwen2.5-VL-72B-Instruct** | 9.6 | 5.9 | 11.9 | 11.1 | 13.6 | 10.5 | / | 54.8 | 56.9 | 56.3 | 56.2 | / |
| **GPT-4o-mini** | 12.3 | 11.9 | 12.5 | 16.7 | 13.6 | 13.4 | / | 59.4 | 60.4 | 59.3 | 59.8 | / |
| **DeepSeek-R1** | 11.7 | 13.9 | 9.5 | 30.6 | 9.1 | 13.9 | / | 63.9 | 61.3 | 61.7 | 62.4 | / |
| *Fine-Tuning Methods with NeuSym RAG* | | | | | | | | | | | | |
| **Qwen2.5-3B-Instruct(SFT)** | 24.7 | 2.0 | 7.5 | 5.6 | 4.5 | 22.2 | 16.7 | 44.8 | 44.5 | 38.8 | 43.3 | 28.4 |
| + M-GRPO | 22.3 | 0.0 | 2.5 | 2.8 | 4.5 | 19.7 | 13.3 | 39.5 | 38.6 | 39.2 | 39.1 | 26.0 |
| + DAPO | 24.3 | **4.0** | 12.5 | 2.8 | **31.8** | 22.4 | 17.1 | 44.6 | 44.7 | 41.7 | 43.9 | 29.3 |
| **Qwen2.5-3B-Instruct(DTFT)** | 25.3 | 2.0 | 7.5 | 5.6 | 4.5 | 22.6 | 17.6 | 41.7 | 41.5 | 38.3 | 40.9 | 30.3 |
| + DFPO(PaperCompass) | 26.6 | 3.0 | 10.0 | 5.6 | 9.1 | 23.7 | 20.0 | 43.6 | 44.2 | 41.9 | 43.5 | 36.1 |
| **Qwen2.5-7B-Instruct(SFT)** | 26.6 | 1.0 | 7.5 | 5.6 | **31.8** | 23.9 | 17.8 | 46.5 | 45.9 | 42.7 | 45.6 | 33.2 |
| + M-GRPO | 24.7 | 1.0 | 0.0 | 0.0 | 9.1 | 21.2 | 16.1 | 46.8 | 47.7 | 41.5 | 46.1 | 32.0 |
| + DAPO | 25.3 | **4.0** | **12.5** | 8.3 | 22.7 | 23.1 | 17.0 | 48.1 | 47.8 | 45.1 | 47.1 | 34.5 |
| **Qwen2.5-7B-Instruct(DTFT)** | **28.5** | 2.0 | 10.0 | 8.3 | 22.7 | **25.5** | 20.2 | 46.9 | 48.4 | 43.3 | 47.0 | 34.9 |
| + DFPO(PaperCompass) | 28.1 | **4.0** | 10.0 | 8.3 | 18.2 | 25.3 | 21.0 | **49.5** | **48.8** | **45.5** | **48.3** | **37.4** |

**Baselines.** We consider the following baselines. ❶*Prompting Method*: To establish strong baselines, we augment general-purpose LLMs (e.g, Qwen2.5, GPT-4, DeepSeek-R1) with Classic-RAG (Lewis et al., 2020). ❷*Fine-Tuning Method*: We instantiate two versions of our PaperCompass using the Qwen2.5-3B-Instruct and Qwen2.5-7B-Instruct[2] as backbones. Both instances are trained following our proposed pipeline, which includes a Draft & Tool-use FT stage. To evaluate the efficacy of our DFPO algorithm, we also create ablation variants where the DFPO component is replaced by two existing RL methods: M-GRPO (Wei et al., 2025) and DAPO (Yu et al., 2025).

**Metrics.** Beyond the default metrics provided with each benchmark, we inspired from (Liu et al., 2025), introducing an additional evaluation metric, I-Avg, defined as:

$$\text{I-Avg} = \text{Avg} \cdot \sqrt{1 - \frac{\bar{I}}{I_{\max}}}, \tag{5}$$

where $\bar{I}$ is the average number of interaction turns, and $I_{\max}$ denotes the maximum number of interaction turns. The I-Avg metric is designed to provide a balanced assessment of agent behavior by jointly considering both its performance and efficiency.

## 4.2 MAIN RESULTS

The main experimental results are reported in Table 1, from which we observe that: ❶**PaperCompass** significantly improves efficiency (I-Avg) without compromising response accuracy (Avg). Specifically, the 3B and 7B versions of our model show an average I-Avg score improvement of 5.5% and 3.2%, respectively, across two evaluated benchmarks. ❷**PaperCompass** demonstrates superior performance in both accuracy and efficiency compared to standard SFT-RL pipelines. For instance, against the DAPO baseline, our 3B and 7B models achieve accuracy gains of 0.5% and 1.7% respectively, while also improving efficiency by 4.9% and 3.5%. In contrast, the M-GRPO baseline even results in performance degradation compared to the SFT-only model, further highlighting the effectiveness of our approach. ❸**PaperCompass** demonstrates competitive performance against proprietary LLM baselines. Notably, our 7B fine-tuned model surpasses the much larger GPT-4o-mini on the complex AirQA-Real benchmark. While the proprietary model's superior general capabilities give it an edge on SciDQA, our results strongly indicate that specialized fine-tuning with strong RAG method like NeuSym-RAG and advanced RL algorithms like DFPO

---

[2]We augment Qwen2.5-3B-Instruct and Qwen2.5-7B-Instruct with NeuSym-RAG (Cao et al., 2025).

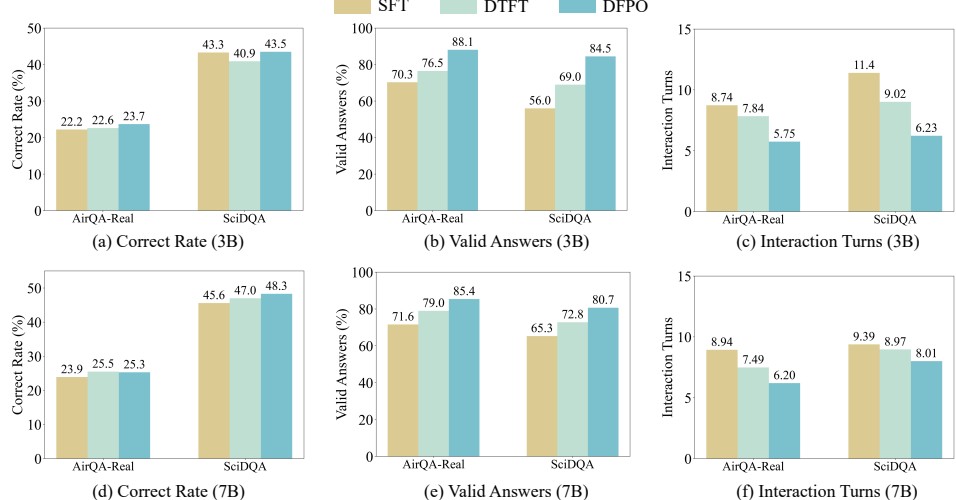

Figure 4: Efficiency statistics of DTFT (Draft & Tool-Use Fine-Tuning) and DFPO compared to SFT baselines on AirQA-Real and SciDQA. The combination of our Draft-and-Follow architecture and the DFPO algorithm enables the agent to operate more efficiently, significantly reducing the required number of tool calls without compromising QA accuracy.

is a highly effective strategy for enabling smaller, open-source models to achieve state-of-the-art performance on domain-specific tasks.

## 4.3 EFFICIENCY STATISTICS

We next investigate how our Draft-and-Follow architecture and DFPO algorithm contribute to improvements in the I-Avg score. To this end, we benchmark our full approach against an SFT-only baseline, with a detailed comparison of their efficiency-related statistics presented in Figure 4.

**Correct Rate.** Our methods, DTFT and DFPO, enhance tool-use efficiency while improving QA correct rate. DTFT yields a 0.3% gain in correct rate over the baseline, with DFPO contributing an additional 1.2%. Both methods leverage a high-quality draft to guide the agent, thereby curtailing ineffective exploration. The resulting accuracy improvement with fewer actions suggests a higher proportion of effective exploration, a conclusion quantitatively supported by the following two metrics.

**Valid Answers.** We measure the valid answers—the proportion of trials concluding within the interaction limit—to quantify the agent's ability to avoid repetitive, non-productive loops. Our methods significantly improve this metric, with DTFT and DFPO yielding increases of 8.8% and 18.9%, respectively. This demonstrates that our draft mechanism effectively prevents agent stagnation, an effect particularly pronounced after applying DFPO.

**Interaction Turns.** Agent efficiency, measured by the average interaction turns, is substantially improved. Our methods, DTFT and DFPO, reduce tool calls by 13.4% and 31.3% from the SFT baseline. This reduction in operational cost, coupled with the aforementioned correct rate gains, confirms a significant enhancement in overall agent performance. We further display the distribution of interaction turns in Appendix A.3.

**Draft-and-Follow and DFPO can narrow the 'knowing-doing' gap.** By externalizing its question understanding and planning into a draft, the agent effectively guides its subsequent exploration. Our experiments demonstrate that an agent guided by this draft adopts a more effective tool-use policy. We interpret this result as a narrowing of the 'knowing-doing' gap. Specifically, the ability to generate a correct draft establishes a solid foundation for 'knowing', while the resulting efficient tool-use demonstrates improved rationality in 'doing'. To further quantify the knowing-doing gap, we use A UCB-based diagnostic bandit task (Schmied et al., 2025) in Appendix A.4.5.

## 4.4 RL METHODS ABLATION

We present a series of ablation studies designed to validate our key architectural choices. First, we investigate the individual contributions of **negative sample masking** (NSM) and the **reward router** (RR) to the performance of PaperCompass. Second, we conduct a direct comparison between our DFPO algorithm and two baseline methods, M-GRPO (Wei et al., 2025) and DAPO (Yu et al., 2025), using the DTFT-trained model as a common starting point.

Based on the quantitative results in Table 2, we draw the following key conclusions and insights (which is detailed in Appendix A.4.3): Our Draft-and-Follow framework significantly improves agent performance by using the draft to guide exploration toward more effective strategies and prevent the reinforcement of spurious reasoning paths. Building on this, our DFPO algorithm achieves superior interaction efficiency by explicitly optimizing the quality of this 'draft' before execution. The framework's success is bolstered by two critical components identified in our ablations: negative sample masking, which prevents destabilizing policy updates from incorrect solutions, and the reward router, which enables a fine-grained distinction between suboptimal and optimal answers.

Table 2: Ablation study on RL method settings. To ensure a fair comparison, all RL method settings are initialized from the same DTFT-trained Qwen-2.5-7B-Instruct checkpoint. The best performance is **bolded**.

| Method | | | AirQA-Real | | SciDQA | |
|---|---|---|---|---|---|---|
| | | | Avg. | I-Avg. | Avg. | I-Avg. |
| **M-GRPO** | | | 24.8 | 18.8 | 45.7 | 34.4 |
| **DAPO** | | | **25.9** | 19.4 | 46.5 | 35.1 |
| | **NSM** | **RR** | | | | |
| **DFPO** | ✗ | ✗ | 23.0 | 18.2 | 45.3 | 34.9 |
| | ✓ | ✗ | 23.8 | 19.1 | 46.2 | 35.3 |
| | ✗ | ✓ | 22.6 | 16.5 | 43.0 | 32.6 |
| | ✓ | ✓ | 25.3 | **21.0** | **48.3** | **37.4** |

## 4.5 ENTROPY ANALYSIS

Recent work has underscored the importance of entropy in LLM-RL (Zhang et al., 2025b; Cui et al., 2025; Xu, 2025). Accordingly, this section presents an analysis of the entropy dynamics observed during the our RL training process. This analysis addresses three key questions: *Why does the M-GRPO baseline underperform compared to the SFT-only model? What is the impact of combining M-GRPO with our DTFT model? From an entropy perspective, how does DFPO further refine the DTFT-trained agent?*

Our analysis of the training dynamics in Figure 5 reveals divergent behaviors across the models. The SFT-based M-GRPO learns a spurious policy; a granular analysis shows its solution entropy paradoxically decreases for incorrect answers while increasing for correct ones, indicating it grows more confident in its errors. When initialized with DTFT, M-GRPO becomes more stable: The draft entropy decreases as expected, but the solution entropy remains high, aligning with the negligible performance gains shown in Figure 1(b). This suggests the draft prevents catastrophic failure but is not further optimized by M-GRPO alone. In sharp contrast, DFPO exhibits the ideal dynamic. Not only are its absolute entropy values lower, but its solution entropy shows the correct trend: Decreasing for correct answers and increasing for incorrect ones. This provides strong evidence that DFPO learns to both pursue correct reasoning paths and actively unlearn erroneous ones.

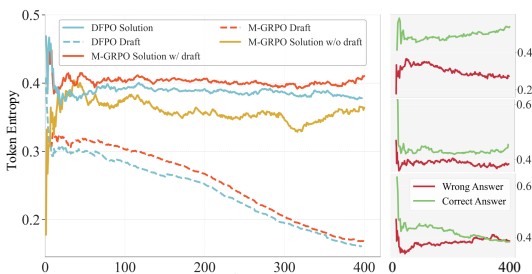

Figure 5: Entropy Dynamics during RL Training. **Left:** Average token entropy curves for the three training settings. **Right:** A granular view of the *solution* entropy, partitioned by correct and wrong final answers. The curves are ordered from top to bottom: M-GRPO w/o draft, M-GRPO w/ draft, and DFPO.

We argue that the assumed correlation between entropy and performance is not applicable in our task, as evidenced by our DFPO analysis, which shows a limited effect on reducing solution entropy. We refute the notion that this is due to insufficient training for two main reasons. First, *the task nature introduces significant differences.* Unlike mathematical reasoning, multi-turn tool use presents uncertainties from variable observations and the model's token distribution, making the relationship between entropy and performance less direct (Dong et al., 2025). Second, *the challenge of*

*deep semantic understanding limits performance gains.* While PaperCompass's accuracy improvement is marginal compared to WebAgent (Wu et al., 2025b), this is due to the inherent complexity of Paper-QA, which demands deep comprehension beyond tool invocation. We emphasize that our main contribution is in improving efficient tool-use strategies, not in enhancing deep semantic understanding, which depends on model scale and pre-training (Rajani et al., 2025; Wu et al., 2025a).

## 5 RELATED WORK

**LLMs for Scientific Research.** The application of LLMs to enhance the scientific research lifecycle is a rapidly expanding field (Luo et al., 2025). The specific directions can be divided into Scientific Hypothesis Discovery (Yang et al., 2023; Wang et al., 2024a;b), Experiment Planning & Implementation (Liu et al., 2023; Rasheed et al., 2024; Schmidgall et al., 2025), Paper Writing (Xing et al., 2020; Wang et al., 2024c; Yu et al., 2024), and Peer Reviewing (Wang et al., 2020; Liu & Shah, 2023; Du et al., 2024). Our work attempts to address Scientific Paper-QA, an emerging research direction where prevailing approaches rely on agents built with Retrieval-Augmented Generation (RAG) (Trivedi et al., 2022; Edge et al., 2024; Sarmah et al., 2024). While the work of Cao et al. (2025) is closely related, our primary contribution is a specialized training process designed to significantly enhance the agent's planning and reasoning abilities.

**Agentic Reinforcement Learning.** A body of recent work has demonstrated the significant potential of Reinforcement Learning (RL) for enhancing the reasoning capabilities of LLMs (Guo et al., 2025; Shen et al., 2025; Yang et al., 2025; Yu et al., 2025). The emergence of tool-integrated reasoning has spurred the application of RL to enhance agent capabilities. These RL-based approaches can be broadly categorized into two main paradigms based on their optimization framework and inference structure: single-layer RL (Feng et al., 2025; Wang et al., 2025b; Li et al., 2025a) and hierarchical RL (Zhou et al., 2024; Hu et al., 2025). Our method, inspired by the options framework (Sutton et al., 1999b), employs a single-layer optimization objective to train a bi-layer inference architecture, aiming to balance the computational efficiency of single-layer methods with the structured reasoning benefits of a hierarchical architecture.

## 6 CONCLUSION

We observed that smaller-scale LLM-based agents tend to get stuck in repetition during retrieval, resulting in severe inefficiency, especially in tasks like Paper-QA that rely on deep semantic understanding. Based on the challenges inherent in Paper-QA and the prevalent 'knowing-doing' gap in LLMs, we introduce **PaperCompass**, a novel agent framework that bridges this divide by guiding subsequent execution ('doing') with a high-level plan ('knowing') in the form of a draft. This approach distinguishes itself from standard SFT-RL pipelines. To effectively support this hierarchical architecture, we developed two specialized methods: Draft & Tool-use Fine-Tuning (DTFT) and Draft-and-Follow Policy Optimization (DFPO). Our experimental results demonstrate that Paper-Compass significantly enhances interaction efficiency without compromising performance.

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

CONTENTS OF APPENDIX

## THE USE OF LARGE LANGUAGE MODELS

Pursuant to the ICLR 2026 policy on Large Language Models (LLMs) usage, we state that LLMs were only used as general writing aids. Specifically, they helped with grammar polishing and readability enhancement. LLMs didn't participate in research ideation, experiment design, data analysis, or result interpretation. Hence, they had no role amounting to authorship or a substantial contribution.

## ETHICS STATEMENT

This work neither involves sensitive personal data nor practices that give rise to privacy or security worries. All datasets employed are publicly accessible and lack personally identifiable information. The research neither puts forward potentially harmful methodologies, applications, nor insights, nor brings up issues associated with discrimination, bias, or fairness. Throughout the research and submission process, the authors have abided by the ICLR Code of Ethics.

## REPRODUCIBILITY STATEMENT

We detail the action space and training formula we designed in Appendix A.1 to Appendix A.3. Additionally, we have made our code publicly available at https://anonymous.4open. science/r/PaperCompass/.

## A  ADDITIONAL DETAILS

### A.1  ACTION SPACE DESIGN

```
RetrieveFromVectorstore(
    # user input can be rephrased
    query: str,
    # select encoding model / modality
    collection_name : str,
    # (table_name, column_name) together defines which view to search
    table_name : str,
    column_name : str,
    # allow fine-grained meta filtering
    filter : str = '',
    limit: int = 5
)
```

```
RetrieveFromDatabase(
    # user input should be recognizable SQL statement
    sql: str
)
```

```
ViewImage(
    # indeed 'pdf_id', obtained from DB
    paper_id : str,
    page_number : int,
    # 4-tuple of float numbers , if [] return the image of entire page
    bounding_box : List [ float ] = []
)
```

```
CalculateExpr(
    # The expression to calculate.
    expr: str
)
```

```
GenerateAnswer(
```

```
2      # The final answer to the user question. Please adhere to the answer
       format for the current question.
3      answer: Any
4  )
```

To ensure stability in our practical implementation, we have made a simplify to `RetrieveFromVectorstore`. The new action, `ClassicRetrieve`, is a variant that retains only the core query and limit parameters.

### A.2 DRAFT & TOOL-USE FINE-TUNING

**Synthetic Expert Trajectory Generation.** To mimic real-world annotation and interaction, the synthesis process is divided into three core components: ❶Explorer constructs a natural language question-answer pair with given context. ❷Tracker chooses suitable evaluation function and fill in the formatted example file. ❸Actor interacts with the outer environment to collect trajectories. We start by downloading 10,000 artificial intelligence papers from arXiv, using tools like PyMuPDF (Artifex Software, Inc., 2023) and MinerU (Wu et al., 2023) to extract metadata, text, and non-textual elements. Following this, the Explorer generates high-quality QA pairs from this content. To improve output quality, this stage incorporates techniques such as chain-of-thought (Wei et al., 2022) and hand-written prompts. First, the process begins with data preparation and the generation of intelligent question-answer (QA) pairs. Next, we use Actor and Tracker components to convert these QA pairs into structured training instructions for an agent. The Actor interacts with the paper database, using the ReAct framework (Yao et al., 2023) and an LLM to simulate a full interaction trajectory of user instructions and agent responses (which include both thought and action). To handle long contexts, these trajectories are processed using a sliding window, and error information is intentionally preserved to train the model's error-correction capabilities. Finally, the Tracker packages these trajectories and QA pairs into formatted training examples. It automatically selects and configures the appropriate evaluation functions, parameters, and answer formats based on the question type, and even uses a rule-based approach to combine simple examples into more complex, multi-part questions. To establish a one-to-one correspondence between each draft and its expert trajectory, we employ a straightforward procedure: an expert LLM (e.g., Qwen2.5-32B-Instruct) is prompted to summarize the trajectory and then reformat the summary into our predefined draft structure. Importantly, to prevent the draft from prematurely revealing key solution steps, it is designed as a high-level plan that is logically derivable from the initial problem description alone.

**Supervised Fine-Tuning with Draft.** $\tau = (u_0, d, y_0 \cdots, u_i, y_i, \cdots, u_N, y_N)$ is an interaction trajectory in a message list manner, where $u_i$ represents the user's instruction, or the observation from the environment, and $y_i$ denotes the response from the agent, including a thought and an action. The policy $\pi_\theta$ is trained via supervised fine-tuning (SFT) with draft, a.k.a. **draft & tool-use fine-tuning (DTFT)** to imitate expert trajectories conditioned on history:

$$\mathcal{L}_{\text{DTFT}} = -\mathbb{E}_{\tau \sim \mathcal{D}} \left[ \underbrace{\log \pi_\theta(d|u_0)}_{\text{Draft}} + \sum_{i=1}^{N} \underbrace{\log \pi_\theta(y_i|h_i)}_{\text{Tool-Use}} \right], \tag{6}$$

where $h_i = (u_0, d, y_0, \cdots, u_{i-1}, y_{i-1}, u_i)$ is the interaction history, a.k.a. the prefix of an expert trajectory.

### A.3 RL TRAINING RECIPE

We adopt a subset of AirQA[3] (Huang et al., 2025) as our training dataset due to its high question diversity and its balanced distribution of difficulty levels. Specifically, to ensure the quality and difficulty of training samples, we apply a special filtering strategy inspired from curriculum learning and hard exapmle mining (Shrivastava et al., 2016; Portelas et al., 2020): We first evaluate a baseline agent (DTFT-trained Qwen2.5-7B-Instruct) on the 693-instance dataset. From this evaluation, we collect the $M = 143$ instances that the agent answered correctly. These instances form the initial seed of our training set. To augment this set, we then randomly sample an additional $400 - M = 257$

---

[3]Since AirQA-Real is a subset of the AirQA dataset, our evaluation uses the set difference which comprises a total of 693 instances.

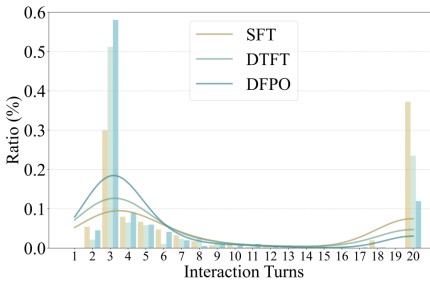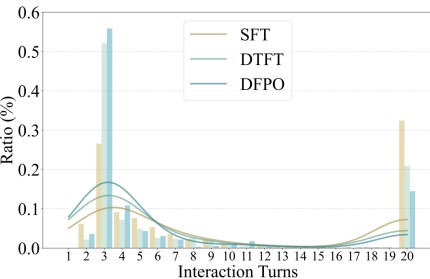

Figure 6: The distribution of interaction turns of the model on AirQA-Real. **Left:** Qwen2.5-3B-Instruct. **Right:** Qwen2.5-7B-Instruct.

instances from the remaining questions, creating a final training set of 400 instances. To avoid introducing any ordering bias during training, the final constructed training set is randomly shuffled.

We adopt the TRL (von Werra et al., 2020) framework for multi-turn RL training, each experiment is trained on 8 Ascend 910B NPUs. We use a cosine learning scheduler from 1e-6 with a batch size of 8. The following is a more detailed hyperparameter report:

- Maximum interaction turn: 10
- Temperature: 0.7
- Top-p: 0.95
- Window size for context: 5
- Action format: Markdown
- Maximum response length (single-turn): 256

### A.4 ADDITIONAL EXPERIMENTS

#### A.4.1 INTERACTION TURNS

While our analysis in Section 4.3 focused on the average number of interaction turns, this section provides a more granular investigation into the full distribution of these turns. Figure 6 and Figure 7 illustrate the distribution of interaction turns for our different training methods. Specifically, the former shows the results on AirQA-Real, while the latter shows the results on SciDQA. On the AirQA-Real, the SFT-only baseline is prone to inefficient exploration. This inefficiency, compounded by the limited semantic understanding of smaller models, often results in repetitive, futile search queries that exhaust the maximum interaction limit. In contrast, the optimal solution path for most tasks involves a concise three-step sequence. Our DTFT and DFPO models successfully learn this efficient strategy, concentrating their interaction turn distributions around this optimal value. A nuanced comparison between our 3B and 7B models reveals further insights. While both models are efficient, the 7B model exhibits a slightly higher average number of interaction turns. This suggests that the larger model leverages its superior comprehension abilities to conduct additional verification steps on more complex questions, rather than prematurely generating a potentially superficial answer. Similar, though less pronounced, trends are observed on the SciDQA. In conclusion, our methods significantly enhance interaction efficiency without compromising performance. Nevertheless, improving the agent's deep semantic comprehension of retrieved content remains a key direction for future research.

#### A.4.2 REPETITION

We next analyze the extent to which PaperCompass exhibits repetitive behavior during training. To quantify this, we first introduce a metric termed the **Repetition Score**. To define the Repetition Score for a single trajectory, we first group all executed actions. Actions are assigned to the same group if and only if their function names and parameters are identical. Let $|\mathcal{C}|$ be the total number of unique action groups, and let $N_i$ be the number of occurrences of the i-th unique action. The

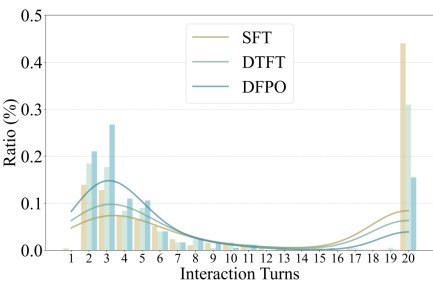 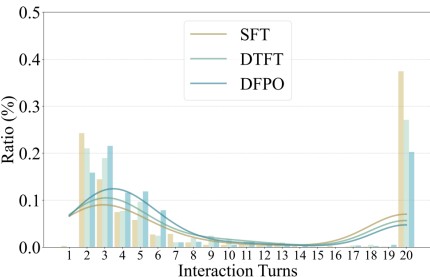

Figure 7: The distribution of interaction turns of the model on SciDQA. **Left:** Qwen2.5-3B-Instruct. **Right:** Qwen2.5-7B-Instruct.

Repetition Score is then formally defined as:

$$\text{Repetion Score} = -0.1 \cdot \left( \max_{j \in \{1, \ldots, |\mathcal{C}|\}} \{N_1, N_2, \ldots, N_{|\mathcal{C}|}\} - 1 \right). \tag{7}$$

As shown in Figure 8, the agent's tendency to repeat actions steadily decreases during training process. We attribute this improvement to two factors. First, by decoupling the optimization of draft and solution, DFPO incentivizes the generation of higher-quality plans. This leads to more effective exploration during the solution phase and reduces the likelihood of the agent entering unproductive loops. Second, we observe a more nuanced recovery strategy. Even when the agent encounters a difficult state, it learns to vary its actions rather than repeating them verbatim. Our training logs reveal that while the agent often re-uses the same retrieval tool, it intelligently modifies the query parameters. This is particularly effective for vector-based methods like `RetrieveFromVectorstore`, where minor parameter changes can yield substantially different results and allow the agent to escape repetitive states. Thus, while the final accuracy gains of PaperCompass may be marginal, the key result is that it learns a significantly more efficient and robust retrieval strategy. We argue that this learned procedural improvement represents the core contribution of our work.

### A.4.3 RL TRAINING SETTING ABLATION

**Draft-and-Follow can increase performance.** When initialized from our DTFT model, both the M-GRPO and DAPO baselines exhibit significantly improved performance compared to their SFT-based counterparts. Notably, the performance degradation previously observed with M-GRPO is mitigated (a finding consistent with Figure 1(b) and Figure 5). This suggests that the draft guides the agent's exploration toward more effective strategies, preventing it from reinforcing spurious reasoning paths that might coincidentally lead to a correct answer.

**DFPO further enhances the interaction efficiency without sacrificing performance.** While DAPO achieves a marginally higher accuracy on AirQA-Real, our DFPO algorithm demonstrates a significant advantage in interaction efficiency. In SciDQA, DFPO achieved SOTA in both of the above metrics. We hypothesize this trade-off stems from differing exploratory behaviors. DAPO's aggressive exploration, driven by its clip-higher strategy, may occasionally allow it to escape local optima. However, without a mechanism to explicitly optimize the draft itself, DAPO can be misled by suboptimal drafts, leading to inefficient search patterns that require many steps to resolve. In contrast, DFPO is explicitly designed

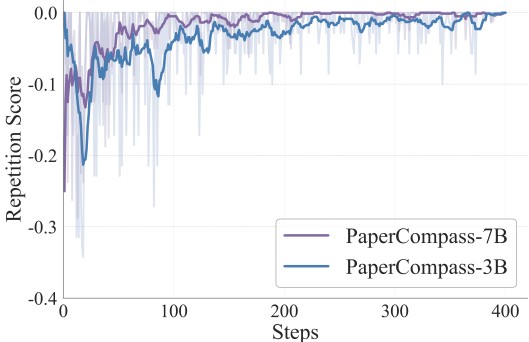

Figure 8: Repetition Score during DFPO training process.

to first optimize the quality of the draft. By ensuring a high-quality plan before execution, DFPO promotes a more focused and efficient exploration strategy from the outset, explaining its superior interaction efficiency.

Table 3: Ablation study on SFT training settings. To ensure fair comparison, all SFT training settings are subsequently trained by DFPO with same configurations.

| Setting | AirQA-Real | | SciDQA | |
|---|---|---|---|---|
| | Avg. | I-Avg. | Avg. | I-Avg. |
| *Qwen2.5-3B-Instruct* | | | | |
| DTFT | 23.7 | 20.0 | 43.5 | 36.1 |
| SFT | 20.4(↓ 13.9%) | 14.8(↓ 26.0%) | 39.5(↓ 9.2%) | 27.7(↓ 23.3%) |
| *Qwen2.5-7B-Instruct* | | | | |
| DTFT | 25.3 | 21.0 | 48.3 | 37.4 |
| SFT | 20.1(↓ 16.6%) | 15.4(↓ 26.7%) | 43.6(↓ 9.7%) | 29.8(↓ 20.3%) |

**Both negative sample masking and reward router are indispensable.** Our ablation study demonstrates that both the negative sample masking and reward router are crucial for performance, with the latter having a more substantial impact. These results align with our theoretical analysis. The removal of negative sample masking is particularly detrimental. As established in Theorem 1, its absence can cause DFPO to erroneously reinforce drafts that lead to incorrect solutions. This introduces conflicting policy gradients, severely destabilizing the optimization process. The reward router addresses a more subtle issue. While a fine-tuned agent is unlikely to engage in simple reward hacking as shown in Appendix C, the router is crucial for reliably differentiating between the quality of various positive samples within a group. Without it, the optimization may incorrectly assign the highest advantage to a suboptimal answer.

### A.4.4 SFT TRAINING SETTING ABLATION

We next investigate whether our proposed DTFT stage is a critical component or if a standard SFT approach would suffice. To this end, we compare two experimental conditions for both the 3B and 7B models: (1) DFPO applied to a DTFT-initialized agent, and (2) DFPO applied to an SFT-initialized agent. The results are shown in Table 3, we observe that agent's performance degrades significantly with a direct SFT. The disproportionately larger performance drop in the I-Avg score, compared to the Avg score, underscores the critical role of the DTFT stage in enhancing the agent's interaction efficiency. We attribute the underperformance of the SFT-based model, relative to the DTFT-initialized version, to the following key factors:

**Incorrect draft format.** Although we provided a one-shot, in-context example of a detailed, step-by-step draft, the agent's planning behavior degraded as training progressed. Initially, the generated draft deviated from the structured format, becoming overly coarse-grained. In the most severe cases, the draft degenerated into a mere repetition or slight rephrasing of the input question, offering no actionable plan.

**Hallucination.** A second failure mode involves the agent attempting to execute tool calls during the draft generation stage. This action is invalid, as no tools are available during this high-level planning phase. More critically, the agent often proceeds to hallucinate a fictitious tool response. This fabricated output severely pollutes the context, leading to irrecoverable errors in the subsequent solution phase.

**Context conflict.** A third failure mode arises during the solution phase, even when the initial draft is correct. At certain forks, the agent can encounter a context conflict where observations from a tool call seem to contradict the high-level plan. The SFT-only agent struggles to arbitrate between these signals, often deviating from the optimal path, it will also cause significant difficulties for the optimization of DFPO. Our DTFT stage is designed to resolve this ambiguity by explicitly training the agent to prioritize and faithfully execute its established draft.

### A.4.5 KNOWING-DOING GAP PROBE

To measure the knowing–doing gap, we use a UCB-based diagnostic bandit task (Schmied et al., 2025). In each episode, the agent is shown 3 arms, each associated with a hidden reward distri-

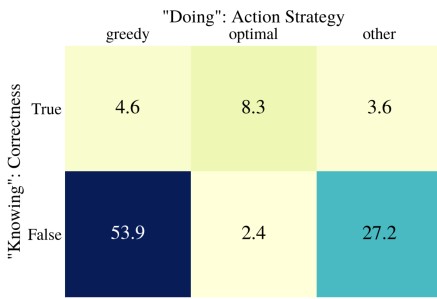 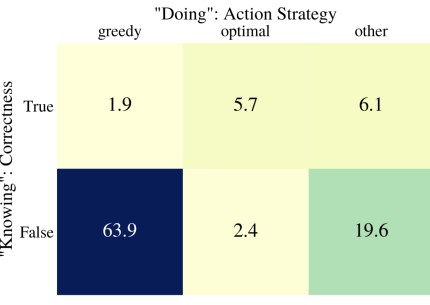

Figure 9: Confusion matrix for the Knowing-Doing Gap of **PaperCompass-7B (Left)** vs. **Qwen2.5-7B-Instruct (Right)**.

bution. The agent must (i) compute the value scores provided in the prompt ("knowing") and (ii) select an arm ("doing"). The optimal arm is uniquely determined from the scores, so a mismatch between reasoning and action directly reflects a knowing–doing gap. We evaluate the model over 64 independent environments, each with 50 decision steps, and report how often the model correctly identifies the optimal arm but fails to choose it. As shown in the Figure 9, PaperCompass-7B not only achieves a higher rate of correct reasoning (16.5% vs. 13.7%), but also conditioned on the steps where the model's reasoning is correct (Knowing=True), it also converts that reasoning into correct actions more reliably: 8.3/16.5 = 50.3% for PaperCompass-7B versus 5.7/13.7 = 41.6% for Qwen2.5-7B-Instruct. This directly indicates a reduction in the knowing–doing gap.

---

**Instructions for PaperCompass-7B or Qwen2.5-7B-Instruct as UCB agent**

Your task is to act according to the Upper-Confidence-Bound (UCB) algorithm. First, briefly write down the UCB algorithm. The UCB value for each arm a at step $t$ is defined as:

$$\mathrm{UCB(a,t)} = \mathrm{avg\_reward(a)} + c \cdot \sqrt{\frac{\log(t)}{\mathrm{pulls(a)}}}$$

where:
- avg_reward(a) is the empirical mean reward of arm a,
- pulls(a) is the number of times arm a has been selected,
- c is a positive exploration coefficient.

Then compute the UCB value for every button (you may approximate the values). Finally, select your action according to the computed UCB quantities.
You MUST output the UCB values in the following format (one per line):

[UCB](If some buttons do not exist in this task, ignore them.)
blue: <value>
green: <value>
red: <value>

Then, at the end of your answer, output your final action in the form:
ACTION=<color>

[History]
So far you have tried/seen:
{history}

Step=$t$ What do you do next?

---

## B  THEORETICAL PROOFS

In this section, we provide detailed proofs of our theoretical results.

### B.1  PROOF FOR PROPOSITION 1

*Proof.* Based on policy gradient theorem (Sutton et al., 1999a), we can derive the policy gradient of DFPO:

$$\nabla_\theta \mathcal{J}_{\text{DFPO}}(\theta) = \mathbb{E}_{q \sim P(q), \{d_i, y_i\}_{i=1}^G \sim (\pi_\theta(d|q), \pi_\theta(y|q,d))}$$

$$\left\{ \frac{1}{G} \sum_{i=1}^G \left[ \frac{1}{|d_i| + |y_i|} \left( \hat{A}_i^{\text{draft}} \nabla_\theta \log \pi_\theta(d_i|q) + \hat{A}_i^{\text{solution}} \nabla_\theta \log \pi_\theta(y_i|q, d_i) \right) \right] \right\}.$$

Since we throw out the constraint of KL divergence, and we utilize rule-based reward as feedback, the advantage should be equal at every moment and is independent of token index.

$$\nabla_\theta \mathcal{J}_{\text{DFPO}}(\theta)$$

$$= \mathbb{E}_{q \sim P(q), \{d_i\}_{i=1}^G \sim \pi_\theta(d|q)} \left\{ \frac{1}{G} \sum_{i=1}^G \left[ \frac{1}{|d_i| + |y_i|} \hat{A}_i^{\text{draft}} \nabla_\theta \log \pi_\theta(d_i|q) \right] \right\}$$

$$+ \mathbb{E}_{q \sim P(q), \{y_i\}_{i=1}^G \sim \pi_\theta(y|q,d)} \left\{ \frac{1}{G} \sum_{i=1}^G \left[ \frac{1}{|d_i| + |y_i|} \hat{A}_i^{\text{solution}} \nabla_\theta \log \pi_\theta(y_i|q, d_i) \right] \right\}$$

$$= \mathbb{E}_{q \sim P(q), \{d_i \circ y_i\}_{i=1}^G \sim \pi_\theta(d \circ y|q)} \left\{ \frac{1}{G} \sum_{i=1}^G \left[ \frac{1}{|d_i| + |y_i|} \hat{A}_i^{\text{solution}} \nabla_\theta \log \pi_\theta(d_i \circ y_i|q) \right] \right\}$$

$$+ \mathbb{E}_{q \sim P(q), \{d_i\}_{i=1}^G \sim \pi_\theta(d|q)} \left\{ \frac{1}{G} \sum_{i=1}^G \left[ \frac{1}{|d_i| + |y_i|} \left( \hat{A}_i^{\text{draft}} - \hat{A}_i^{\text{solution}} \right) \nabla_\theta \log \pi_\theta(d_i|q) \right] \right\}$$

$$= \nabla_\theta \mathcal{J}_{\text{M-GRPO}}(\theta)$$

$$+ \mathbb{E}_{q \sim P(q), \{d_i\}_{i=1}^G \sim \pi_\theta(d|q)} \left\{ \frac{1}{G} \sum_{i=1}^G \left[ \frac{1}{|d_i| + |y_i|} \left( \hat{A}_i^{\text{draft}} - \hat{A}_i^{\text{solution}} \right) \nabla_\theta \log \pi_\theta(d_i|q) \right] \right\}$$

$$= \nabla_\theta \mathcal{J}_{\text{M-GRPO}}(\theta) + \mathbb{E}_{q \sim P(q), \{d_i\}_{i=1}^G \sim \pi_\theta(d|q)} \left\{ \frac{1}{G} \sum_{i=1}^G \left[ \Delta \hat{A}_i \nabla_\theta \log \pi_\theta(d_i|q) \right] \right\}.$$

$$\square$$

Considering a more general setting, we now analyze rollouts generated from old policy—a technique central to algorithms like PPO (Schulman et al., 2017) and GRPO (Shao et al., 2024). Then, the objective function of DFPO, which we call as DFPO-off, becomes the following form:

$$\mathcal{J}_{\text{DFPO-off}}(\theta) = \mathbb{E}_{q \sim P(q), \{d_i, y_i\}_{i=1}^G \sim (\pi_{\text{old}}(d|q), \pi_{\text{old}}(y|q,d))}$$

$$\left\{ \frac{1}{G} \sum_{i=1}^G \left[ \frac{1}{|d_i| + |y_i|} \left( \sum_{t=1}^{|d_i|} \rho_{i,t}^{\text{draft}} + \sum_{t=|d_i|+1}^{|d_i|+|y_i|} \rho_{i,t}^{\text{solution}} \right) \right] \right\}, \tag{8}$$

where $\rho$ denotes the clipping operation:

$$\begin{cases} \rho_{i,t}^{\text{draft}} = \min \left\{ \frac{\pi_\theta(d_{i,t}|q, d_{i,<t})}{\pi_{\text{old}}(d_{i,t}|q, d_{i,<t})} \hat{A}_{i,t}^{\text{draft}}, \text{clip} \left( \frac{\pi_\theta(d_{i,t}|q, d_{i,<t})}{\pi_{\text{old}}(d_{i,t}|q, d_{i,<t})}, 1 - \epsilon, 1 + \epsilon \right) \hat{A}_{i,t}^{\text{draft}} \right\} \\ \rho_{i,t}^{\text{solution}} = \min \left\{ \frac{\pi_\theta(y_{i,t}|q, d_i, y_{i,<t})}{\pi_{\text{old}}(y_{i,t}|q, d_i, y_{i,<t})} \hat{A}_{i,t}^{\text{solution}}, \text{clip} \left( \frac{\pi_\theta(y_{i,t}|q, d_i, y_{i,<t})}{\pi_{\text{old}}(y_{i,t}|q, d_i, y_{i,<t})}, 1 - \epsilon, 1 + \epsilon \right) \hat{A}_{i,t}^{\text{solution}} \right\} \end{cases}$$

We analyze the signs of the draft and solution advantages, assuming the application of negative sample masking. As mentioned in Section 3.3, we directly set $\hat{A}_{i,t}^{\text{draft}} = \hat{A}_i^{\text{draft}}$ and $\hat{A}_{i,t}^{\text{solution}} = \hat{A}_i^{\text{solution}}$.

**Case 1:** If solution is incorrect, we get $\hat{A}_{i,t}^{\text{draft}} = \hat{A}_{i,t}^{\text{solution}} \leq 0$,

$$\min\left\{\frac{\pi_\theta(d_{i,t}|q, d_{i,<t})}{\pi_{\text{old}}(d_{i,t}|q, d_{i,<t})}\hat{A}_i^{\text{draft}}, \text{clip}\left(\frac{\pi_\theta(d_{i,t}|q, d_{i,<t})}{\pi_{\text{old}}(d_{i,t}|q, d_{i,<t})}, 1-\epsilon, 1+\epsilon\right)\hat{A}_i^{\text{draft}}\right\}$$

$$= \min\left\{\frac{\pi_\theta(d_{i,t}|q, d_{i,<t})}{\pi_{\text{old}}(d_{i,t}|q, d_{i,<t})}\hat{A}_i^{\text{solution}}, \text{clip}\left(\frac{\pi_\theta(d_{i,t}|q, d_{i,<t})}{\pi_{\text{old}}(d_{i,t}|q, d_{i,<t})}, 1-\epsilon, 1+\epsilon\right)\hat{A}_i^{\text{solution}}\right\}.$$

**Case 2:** If solution is correct, we get $\hat{A}_{i,t}^{\text{solution}} \geq 0$,

**Case 2.1:** If draft reward is relative high, we get $\hat{A}_{i,t}^{\text{draft}} \geq 0$,

$$\min\left\{\frac{\pi_\theta(d_{i,t}|q, d_{i,<t})}{\pi_{\text{old}}(d_{i,t}|q, d_{i,<t})}\hat{A}_i^{\text{draft}}, \text{clip}\left(\frac{\pi_\theta(d_{i,t}|q, d_{i,<t})}{\pi_{\text{old}}(d_{i,t}|q, d_{i,<t})}, 1-\epsilon, 1+\epsilon\right)\hat{A}_i^{\text{draft}}\right\}$$

$$= \min\left\{\frac{\pi_\theta(d_{i,t}|q, d_{i,<t})}{\pi_{\text{old}}(d_{i,t}|q, d_{i,<t})}, \text{clip}\left(\frac{\pi_\theta(d_{i,t}|q, d_{i,<t})}{\pi_{\text{old}}(d_{i,t}|q, d_{i,<t})}, 1-\epsilon, 1+\epsilon\right)\right\}\hat{A}_i^{\text{draft}}$$

$$= \min\left\{\frac{\pi_\theta(d_{i,t}|q, d_{i,<t})}{\pi_{\text{old}}(d_{i,t}|q, d_{i,<t})}, \text{clip}\left(\frac{\pi_\theta(d_{i,t}|q, d_{i,<t})}{\pi_{\text{old}}(d_{i,t}|q, d_{i,<t})}, 1-\epsilon, 1+\epsilon\right)\right\}\hat{A}_i^{\text{solution}}$$

$$+ \min\left\{\frac{\pi_\theta(d_{i,t}|q, d_{i,<t})}{\pi_{\text{old}}(d_{i,t}|q, d_{i,<t})}, \text{clip}\left(\frac{\pi_\theta(d_{i,t}|q, d_{i,<t})}{\pi_{\text{old}}(d_{i,t}|q, d_{i,<t})}, 1-\epsilon, 1+\epsilon\right)\right\}\left(\hat{A}_i^{\text{draft}} - \hat{A}_i^{\text{solution}}\right)$$

**Case 2.2:** If draft reward is relative low, we get $\hat{A}_{i,t}^{\text{draft}} \leq 0$,

$$\min\left\{\frac{\pi_\theta(d_{i,t}|q, d_{i,<t})}{\pi_{\text{old}}(d_{i,t}|q, d_{i,<t})}\hat{A}_i^{\text{draft}}, \text{clip}\left(\frac{\pi_\theta(d_{i,t}|q, d_{i,<t})}{\pi_{\text{old}}(d_{i,t}|q, d_{i,<t})}, 1-\epsilon, 1+\epsilon\right)\hat{A}_i^{\text{draft}}\right\}$$

$$= \max\left\{\frac{\pi_\theta(d_{i,t}|q, d_{i,<t})}{\pi_{\text{old}}(d_{i,t}|q, d_{i,<t})}, \text{clip}\left(\frac{\pi_\theta(d_{i,t}|q, d_{i,<t})}{\pi_{\text{old}}(d_{i,t}|q, d_{i,<t})}, 1-\epsilon, 1+\epsilon\right)\right\}\hat{A}_i^{\text{draft}}$$

$$\leq \min\left\{\frac{\pi_\theta(d_{i,t}|q, d_{i,<t})}{\pi_{\text{old}}(d_{i,t}|q, d_{i,<t})}\hat{A}_i^{\text{solution}}, \text{clip}\left(\frac{\pi_\theta(d_{i,t}|q, d_{i,<t})}{\pi_{\text{old}}(d_{i,t}|q, d_{i,<t})}, 1-\epsilon, 1+\epsilon\right)\hat{A}_i^{\text{solution}}\right\}$$

Then, we derive the surrogate objective function as follow:

$$\mathcal{J}_{\text{surrogate}}(\theta) = \mathbb{E}_{q \sim P(q), \{d_i, y_i\}_{i=1}^G \sim (\pi_{\text{old}}(d|q), \pi_{\text{old}}(y|q,d))}$$

$$\frac{1}{G}\left\{\sum_{i=1}^{G}\left[\frac{1}{|d_i|+|y_i|}\sum_{t=1}^{|d_i|+|y_i|}\rho_{i,t}^{\text{draft-solution}}\right] + \sum_{j=1}^{G'}\left[\frac{1}{|d_j|+|y_j|}\sum_{t=1}^{|d_j|}R_{j,t}\left(\hat{A}_j^{\text{draft}} - \hat{A}_j^{\text{solution}}\right)\right]\right\},$$

where $G'$ is the case number of case 2.1,

$$\sum_{t=1}^{|d_i|+|y_i|}\rho_{i,t}^{\text{draft-solution}} = \sum_{t=1}^{|d_i|}\min\left\{\frac{\pi_\theta(d_{i,t}|q, d_{i,<t})}{\pi_{\text{old}}(d_{i,t}|q, d_{i,<t})}\hat{A}_i^{\text{solution}}, \text{clip}\left(\frac{\pi_\theta(d_{i,t}|q, d_{i,<t})}{\pi_{\text{old}}(d_{i,t}|q, d_{i,<t})}, 1-\epsilon, 1+\epsilon\right)\hat{A}_i^{\text{solution}}\right\}$$

$$+ \sum_{t=1}^{|d_i|}\min\left\{\frac{\pi_\theta(y_{i,t}|q, d_i, y_{i,<t})}{\pi_{\text{old}}(y_{i,t}|q, d_i, y_{i,<t})}\hat{A}_{i,t}^{\text{solution}}, \text{clip}\left(\frac{\pi_\theta(y_{i,t}|q, d_i, y_{i,<t})}{\pi_{\text{old}}(y_{i,t}|q, d_i, y_{i,<t})}, 1-\epsilon, 1+\epsilon\right)\hat{A}_{i,t}^{\text{solution}}\right\},$$

and

$$R_{j,t} = \min\left\{\frac{\pi_\theta(d_{j,t}|q, d_{j,<t})}{\pi_{\text{old}}(d_{j,t}|q, d_{j,<t})}, \text{clip}\left(\frac{\pi_\theta(d_{j,t}|q, d_{j,<t})}{\pi_{\text{old}}(d_{j,t}|q, d_{j,<t})}, 1-\epsilon, 1+\epsilon\right)\right\}.$$

Through algebraic manipulation, the expression simplifies to:

$$\mathcal{J}_{\text{DFPO-off}}(\theta) \leq \mathcal{J}_{\text{surrogate}}(\theta)$$

$$= \mathcal{J}_{\text{M-GRPO-off}}(\theta) + \mathbb{E}\left\{\sum_{j=1}^{G'}\left[\frac{1}{|d_j|+|y_j|}\sum_{t=1}^{|d_j|}R_{j,t}\left(\hat{A}_j^{\text{draft}} - \hat{A}_j^{\text{solution}}\right)\right]\right\}. \quad (9)$$

Maximizing the DFPO-off objective, therefore, optimizes a lower bound on the surrogate objective. We've also found that the relationship between this surrogate and the M-GRPO-off policy gradient is analogous to what we established in Proposition 1. This is a significant result, as it shows the theoretical guarantees of Proposition 1 extend to the near off-policy setting.

## B.2 PROOF FOR THEOREM 1

To prove Theorem 1, we first establish a definition and two essential lemmas before proving the main theorem.

**Definition 1** (Index Partitions). *We partition the set of indices $\{1, \ldots, n\}$ into two disjoint sets:*

- *$I_1 = \{i \mid r_i^{solution} = 1\}$, with cardinality $n_1 = |I_1|$.*

- *$I_0 = \{i \mid r_i^{solution} = 0\}$, with cardinality $n_0 = |I_0|$.*

*We assume $n_1 > 0$ and $n_0 > 0$ to ensure non-trivial cases.*

**Lemma 1** (Coefficient of Variation Inequality). *The coefficients of variation for $\boldsymbol{r}^{solution}$ and $\boldsymbol{r}^{draft}$ satisfy the inequality:*

$$\mathrm{CV}(\mathbf{r}^{\mathrm{draft}}) \geq \mathrm{CV}(\mathbf{r}^{\mathrm{solution}}). \tag{10}$$

*Proof.* It is sufficient to prove the inequality for the squared values, $\mathrm{CV}(\boldsymbol{r}^{\mathrm{draft}})^2 \geq \mathrm{CV}(\boldsymbol{r}^{\mathrm{solution}})^2$. For $\boldsymbol{r}^{\mathrm{solution}}$, we have $\bar{r}^{\mathrm{solution}} = n_1/n$ and $s_{r^{\mathrm{solution}}}^2 = n_1 n_0/n^2$. Thus,

$$\mathrm{CV}(\boldsymbol{r}^{\mathrm{solution}})^2 = \frac{s_{r^{\mathrm{solution}}}^2}{(\bar{r}^{\mathrm{solution}})^2} = \frac{n_1 n_0/n^2}{(n_1/n)^2} = \frac{n_0}{n_1}.$$

For $\boldsymbol{r}^{\mathrm{draft}}$, let:

$$\mu_{d1} = \frac{1}{n_1} \sum_{j \in I_1} d_j \quad \text{and} \quad \sigma_{d1}^2 = \frac{1}{n_1} \sum_{j \in I_1} (d_j - \mu_{d1})^2.$$

It can be shown that $\bar{r}^{\mathrm{draft}} = \frac{n_1}{n} \mu_{d1}$ and $s_{r^{\mathrm{draft}}}^2 = \frac{n_1}{n} \sigma_{d1}^2 + \frac{n_1 n_0}{n^2} \mu_{d1}^2$. This gives:

$$\mathrm{CV}(\boldsymbol{r}^{\mathrm{draft}})^2 = \frac{s_{r^{\mathrm{draft}}}^2}{(\bar{r}^{\mathrm{draft}})^2} = \frac{\frac{n_1}{n} \sigma_{d1}^2 + \frac{n_1 n_0}{n^2} \mu_{d1}^2}{(\frac{n_1}{n} \mu_{d1})^2} = \frac{n}{n_1} \frac{\sigma_{d1}^2}{\mu_{d1}^2} + \frac{n_0}{n_1}.$$

The inequality $\mathrm{CV}(\boldsymbol{r}^{\mathrm{draft}})^2 \geq \mathrm{CV}(\boldsymbol{r}^{\mathrm{solution}})^2$ becomes

$$\frac{n}{n_1} \frac{\sigma_{d1}^2}{\mu_{d1}^2} + \frac{n_0}{n_1} \geq \frac{n_0}{n_1},$$

this holds true since variance $\sigma_{d1}^2 \geq 0$. $\qquad\square$

**Lemma 2** (Variance Bound for Bounded Data, Bhatia & Davis (2000)). *Let $\boldsymbol{r}^{draft}$ be a vector with all values in $[0, r_{i_{max}}^{draft}]$. Its population variance $s_{r^{draft}}^2$ is bounded by $s_{r^{draft}}^2 \leq \bar{r}^{draft}(r_{i_{max}}^{draft} - \bar{r}^{draft})$.*

*Proof.* From assumption, it is evident that both the term $(r_i^{\mathrm{draft}} - 0)$ and the term $(r_{i_{\max}}^{\mathrm{draft}} - r_i^{\mathrm{draft}})$ are non-negative. Therefore, their product must also be non-negative:

$$(r_{i_{\max}}^{\mathrm{draft}} - r_i^{\mathrm{draft}})(r_i^{\mathrm{draft}} - 0) \geq 0.$$

This inequality holds for every element $i = 1, \ldots, n$. We can therefore sum this expression over all $n$ data points, and the resulting sum will also be non-negative:

$$\sum_{i=1}^{n} \left( r_{i_{\max}}^{\mathrm{draft}} \cdot r_i^{\mathrm{draft}} - (r_i^{\mathrm{draft}})^2 \right) \geq 0.$$

By the linearity of summation, we can distribute the sum:

$$r_{i_{\max}}^{\mathrm{draft}} \sum_{i=1}^{n} r_i^{\mathrm{draft}} - \sum_{i=1}^{n} (r_i^{\mathrm{draft}})^2 \geq 0.$$

To introduce the mean ($\bar{r}^{\mathrm{draft}}$) and variance ($s_{r^{\mathrm{draft}}}^2$), we divide the entire inequality by $n$:

$$r_{i_{\max}}^{\mathrm{draft}} \left( \frac{1}{n} \sum_{i=1}^{n} r_i^{\mathrm{draft}} \right) - \left( \frac{1}{n} \sum_{i=1}^{n} (r_i^{\mathrm{draft}})^2 \right) \geq 0.$$

We know that the population variance is defined as $s_{r^{\text{draft}}}^2 = \left(\frac{1}{n}\sum_{i=1}^n (r_i^{\text{draft}})^2\right) - (\bar{r}^{\text{draft}})^2$. This can be rearranged to $\left(\frac{1}{n}\sum_{i=1}^n (r_i^{\text{draft}})^2\right) = s_{r^{\text{draft}}}^2 + (\bar{r}^{\text{draft}})^2$. Substituting this into our inequality gives:

$$r_{i_{\max}}^{\text{draft}} \cdot \bar{r}^{\text{draft}} - (s_{r^{\text{draft}}}^2 + (\bar{r}^{\text{draft}})^2) \geq 0.$$

Finally, we rearrange the terms to isolate the variance $s_{r^{\text{draft}}}^2$:

$$s_{r^{\text{draft}}}^2 \leq r_{i_{\max}}^{\text{draft}} \cdot \bar{r}^{\text{draft}} - (\bar{r}^{\text{draft}})^2.$$

Factoring the right-hand side gives the desired result:

$$s_{r^{\text{draft}}}^2 \leq \bar{r}^{\text{draft}}(r_{i_{\max}}^{\text{draft}} - \bar{r}^{\text{draft}}).$$

$\square$

Next, we formally prove the Theorem 1. To make our proofs briefly, with a slight abuse of symbols, all the advantages that appear in the following proofs are the original advantage that is before evenly distributed each token.

*Proof.* **Part 1:** For any $i \in I_0$, $r_i^{\text{solution}} = r_i^{\text{draft}} = 0$. The advantages are

$$\hat{A}_i^{\text{solution}} = -\frac{1}{\text{CV}(\boldsymbol{r}^{\text{solution}})} \quad \text{and} \quad \hat{A}_i^{\text{draft}} = -\frac{1}{\text{CV}(\boldsymbol{r}^{\text{draft}})}.$$

The inequality $\hat{A}_i^{\text{solution}} \leq \hat{A}_i^{\text{draft}}$ is thus equivalent to $\text{CV}(\boldsymbol{r}^{\text{draft}}) \geq \text{CV}(\boldsymbol{r}^{\text{solution}})$. This is proven by Lemma 1.

**Part 2:** For any $i \in I_1$, there is

$$r_i^{\text{solution}} = \bar{r}_{I_1}^{\text{solution}} \geq \frac{|I_1|}{|I_0| + |I_1|} \cdot \bar{r}_{I_1}^{\text{solution}} = \bar{r}^{\text{solution}},$$

which indicates that $\hat{A}_i^{\text{solution}} \geq 0$. The inequality $\hat{A}_i^{\text{draft}} \leq \hat{A}_i^{\text{solution}}$ is thus equivalent to $r_i^{\text{draft}} \leq \bar{r}^{\text{draft}}$, assumed below.

**Part 3:** For $i_{\max} \in I_1$, $\hat{A}_{i_{\max}}^{\text{solution}} = (1 - \bar{r}^{\text{solution}})(s_{r^{\text{solution}}}) = \sqrt{n_0/n_1}$. We need to prove $\sqrt{n_0/n_1} \leq \hat{A}_{i_{\max}}^{\text{draft}}$. Squaring both sides yields the equivalent inequality:

$$\frac{n_0}{n_1} \leq (\hat{A}_{i_{\max}}^{\text{draft}})^2 = \frac{(r_{i_{\max}}^{\text{draft}} - \bar{r}^{\text{draft}})^2}{s_{r^{\text{draft}}}^2}.$$

We apply Lemma 2, which gives $s_{r^{\text{draft}}}^2 \leq \bar{r}^{\text{draft}}(r_{i_{\max}}^{\text{draft}} - \bar{r}^{\text{draft}})$. This implies a lower bound on the squared advantage:

$$(\hat{A}_{i_{\max}}^{\text{draft}})^2 \geq \frac{(r_{i_{\max}}^{\text{draft}} - \bar{r}^{\text{draft}})^2}{\bar{r}^{\text{draft}}(r_{i_{\max}}^{\text{draft}} - \bar{r}^{\text{draft}})} = \frac{r_{i_{\max}}^{\text{draft}}}{\bar{r}^{\text{draft}}} - 1.$$

The theorem holds if this lower bound is greater than or equal to $n_0/n_1$:

$$\frac{r_{i_{\max}}^{\text{draft}}}{\bar{r}^{\text{draft}}} - 1 \geq \frac{n_0}{n_1} \iff \frac{r_{i_{\max}}^{\text{draft}}}{\bar{r}^{\text{draft}}} \geq \frac{n_0 + n_1}{n_1} = \frac{n}{n_1}.$$

Substituting $\bar{r}^{\text{draft}} = \frac{1}{n}\sum_{j \in I_1} r_j^{\text{draft}}$, this simplifies to:

$$r_{i_{\max}}^{\text{draft}} \geq \frac{1}{n_1} \sum_{j \in I_1} r_j^{\text{draft}}.$$

This final inequality is true, as the maximum of a set is always greater than or equal to its mean. $\square$

## C    REWARD ROUTER

Existing works in Agentic RL (Chen et al., 2025; Zheng et al., 2025; Jin et al., 2025) have often utilized dense outcome rewards based on lexical overlap metrics like the F1-score. This approach is favored for its computational simplicity and independence from manual annotation. However, when applied to the domain of scientific papers, our analysis indicates that this reliance on surface-level text matching exhibits significant limitations. This shortcoming is illustrated by the following example:

---

**An Example for F1 Score**

**Golden Answer:** The proposed focus on taking a significant step forward in learning high-performance generalist agents.

**Predicted Answer 1:** This paper concentrates on taking a important step forward in learning outperformed generalist agents.
**Predicted Answer 2:** The proposed focus on taking a significant step forward in learning high-performance expert agents.
**Predicted Answer 3:** In learning high-performance generalist agents In learning high-performance generalist agents In learning high-performance generalist agents.

---

From a semantic standpoint, Predicted Answer 1 is clearly the most appropriate response. However, calculating the F1-score for each predicted answer leads to a different conclusion:

$$\text{F1-Score} = \frac{2 * \text{IN}}{\text{PN} + \text{GN}}.$$

Here, PN denotes the number of tokens in the predicted answer, GN denotes the number of tokens in the golden answer, and IN represents the number of overlapping tokens between the two. This yields F1 scores of 0.643, 0.929, and 0.357 for the three predicted answers, respectively. This example illustrates the significant drawbacks of using the F1 score directly as a reward signal:

- **Failure to Distinguish Between High-Quality and Low-Quality Answers:** Predicted Answer 2, which is factually incorrect and a result of agent hallucination, nevertheless receives a very high reward. Conversely, Predicted Answer 1, a semantically equivalent paraphrase of the golden answer, is assigned a comparatively low score.

- **Susceptibility to Reward Hacking:** Predicted Answer 3 consists of nonsensical repetitions, yet it still manages to obtain a non-trivial score. This demonstrates that an agent can easily exploit the F1 metric to gain rewards for meaningless outputs.

To account for diverse question types, we introduce a reward router that provides a more accurate evaluation of the agent's output quality, thus yielding higher-fidelity gradient signals. Reward router selects an appropriate reward function for each problem type from a predefined suite of 17 evaluation functions based on Cao et al. (2025). In cases where the chosen function yields a continuous score, a threshold is applied to binarize the output for use as a final reward signal.

## D    TRAJECTORY ENTROPY & NEGATIVE SAMPLE MASKING

Our implementation of PaperCompass incorporates two key techniques: Trajectory entropy (Agarwal et al., 2025) as a quality metric and negative sample masking to discourage the generation of low-quality drafts. This section presents a qualitative analysis to investigate the impact of these components on agent behavior.

Table 4: The checklist of the 17 used evaluation functions, including their categories, names, and descriptions.

| Eval Type | Sub-Type | Function | Description |
|---|---|---|---|
| objective | match | eval_bool_exact_match | Evaluate the output against the answer using exact boolean match. |
| | | eval_float_exact_match | Evaluate the output against the answer using exact float match with variable precision or tolerance. |
| | | eval_int_exact_match | Evaluate the output against the answer using exact integer match. |
| | | eval_string_exact_match | Evaluate the output against the answer using exact string match. |
| | | eval_structured_object _exact_match | Evaluate the output against the answer recursively by parsing them both as Python-style lists or dictionaries. |
| | set | eval_element_included | Evaluate whether the output is included in the answer list. |
| | | eval_element_list_included | Evaluate whether each element in the output list is included in the answer list. |
| | | eval_element_list_overlap | Evaluate whether the output list overlaps with the answer list. |
| | retrieval | eval_paper_relevance_with _reference_answer | Evaluate whether the retrieved paper is the same as the reference answer. |
| subjective | semantic | eval_reference_answer _with_llm | Evaluate the output against the reference answer using LLMs. |
| | | eval_scoring_points_with _llm | Evaluate whether the scoring points are all mentioned in the output using LLMs. |
| | | eval_partial_scoring_points _with_llm | Evaluate whether the scoring points are partially mentioned in the output using LLMs. |
| | formula | eval_complex_math_form ula_with_llm | Evaluate the mathematical equivalence between the output and the answer formatted in Latex using LLMs. |
| logical | | eval_conjunction | Evaluate the conjunction of multiple evaluation functions. The output passes the evaluation if and only if all the elements in the output pass the corresponding sub-evaluations. |
| | | eval_disjunction | Evaluate the disjunction of multiple evaluation functions. The output passes the evaluation if and only if at least one of the element in the output passes the corresponding sub-evaluation. |
| | | eval_negation | Evaluate the negation of an evaluation function. The output passes the evaluation if and only if it doesn't pass the original evaluation function. |
| others | | eval_scidqa | Evaluate examples in dataset SciDQA with the encapsulated original LLM-based function. |

## D.1 POLICY UPDATE

**Proposition 2** (Stochastic Parameter Update for Tabular Softmax Policy). *Let $\pi_\theta(a|s)$ be a tabular softmax policy parameterized by $\theta_{s,a} \in \mathbb{R}$ for each state-action pair $(s, a)$. The policy is defined as:*

$$\pi_\theta(a|s) = \frac{\exp(\theta_{s,a})}{\sum_{b \in \mathcal{A}(s)} \exp(\theta_{s,b})}$$

*Consider a stochastic gradient ascent update at step $k$ on the policy gradient objective, using a learning rate $\alpha > 0$. The update is based on a single transition experience $(s_t, a_t)$ and an associated advantage estimate $A^k(s_t, a_t)$. The resulting change in the parameters, $\Delta\theta_{s_t,a'} := \theta^{k+1}_{s_t,a'} - \theta^k_{s_t,a'}$, for any action $a' \in \mathcal{A}(s_t)$ is given by:*

$$\Delta\theta_{s_t,a'} = \begin{cases} \alpha \left(1 - \pi^k_\theta(a_t|s_t)\right) A^k(s_t, a_t), & \text{if } a' = a_t \\ -\alpha \, \pi^k_\theta(a'|s_t) A^k(s_t, a_t), & \text{if } a' \neq a_t \end{cases} \quad (11)$$

Focusing on the draft generation process, the application of negative sample masking modifies the draft advantage term from Proposition 2 as follows:

$$A^k(s_t, a_t) = \hat{A}^{\text{draft}}_i = \begin{cases} \hat{A}^{\text{solution}}_i, & \text{if } r^{\text{solution}}_i = 0 \\ \dfrac{\rho(d_i) - \text{mean}_j(\rho(d_j))}{\text{std}_j(\rho(d_j))}, & \text{if } r^{\text{solution}}_i = 1 \end{cases}$$

Where $-\rho(\cdot)$ is trajectory entropy. To align the settings with those used in PaperCompass, we analyze the parameter update direction under the assumption that *negative sample masking is applied*. It can be divided into the following situations:

❶**Correct solution; Low trajectory entropy.** As a result of this update, the new policy assigns a higher probability to generating the target draft and, conversely, a lower probability to other potential drafts. This process establishes a positive feedback loop: drafts that are both high-quality and generated with high confidence receive a substantial reward, the magnitude of which is scaled by that confidence. This, in turn, reinforces the policy, increasing the likelihood that similar effective drafts will be generated in the future.

❷**Wrong solution; Low trajectory entropy.** For a wrong solution, our advantage calculation leads to $\hat{A}^{\text{draft}}_i = \hat{A}^{\text{solution}}_i \leq 0$. If the group contains a correct solution, the relative advantage term becomes negative, and the policy update consequently suppresses the associated low-quality draft, thereby promoting exploration. Moreover, negative sample masking addresses the worst-case scenario: if a group consists entirely of negative samples, this mechanism prevents any policy update, thus avoiding the erroneous reinforcement of poor-quality drafts.

❸**Correct solution; High trajectory entropy.** In contrast to ❶, a draft that leads to a correct solution but is generated with low confidence (i.e., high trajectory entropy) is penalized. The policy update suppresses the probability of this particular draft, redistributing the likelihood across other potential plans. The rationale for this counter-intuitive mechanism is to discourage the agent from relying on overly general or lucky guess, which might succeed on simple quesions but do not reflect a robust understanding of the task. Ultimately, the goal is to incentivize the generation of drafts that are not only effective but also well-reasoned and produced with high confidence.

❹**Wrong solution; High trajectory entropy.** Similar to ❷, the advantage for such a draft is guaranteed to be non-positive under negative sample masking. Given that the draft was generated with low confidence, the magnitude of the resulting suppressive policy gradient is amplified.

## D.2 POLICY ENTROPY

**Lemma 3** (Policy Entropy Update, Zhang et al. (2025b)). *Let the actor policy $\pi_\theta$ be a tabular softmax policy, the difference of information entropy $\mathcal{H}$ given state $s$ between two consecutive steps satisfies*

$$\mathcal{H}(\pi^{k+1}_\theta|s) - \mathcal{H}(\pi^k_\theta|s) \approx -Cov_{a \sim \pi^k_\theta(\cdot|s)} \left(\log \pi^k_\theta(a|s), \, \Delta\theta_{s,a}\right). \quad (12)$$

According to our previous analysis, the policy we concern is the likelihood of draft, that is, $\pi_\theta^k(a|s) = \pi_\theta^k(d_{i,t}|q, d_{i,<t})$. Combining the results from Appendix D.1 and Lemma 3, we observe that the policy entropy decreases only when the agent generates a high-quality 'draft' with high confidence (encouraging exploitation), and increases in all other cases (promoting exploration). Our experiments further confirm that although the entropy of successful drafts decreases, the policy does not suffer from mode collapse. This behavior reflects the intended role of our technical choices: in particular, the external verifiable signal and negative sample masking, which is introduced to prevent reinforcing erroneous drafts and thereby stabilize the optimization dynamics. They differentiate our framework from many Reinforcement Learning from Internal Feedback (RLIF) methods (Agarwal et al., 2025; Zhao et al., 2025).

### D.3 Sketch Proofs

*Proof.* With the policy gradient theorem (Sutton et al., 1999a; Shao et al., 2024), which states that the gradient of the objective function $\mathcal{J}(\theta)$ can be expressed using the advantage function $A^\pi(s, a)$ as:

$$\nabla_\theta \mathcal{J}(\theta) = \mathbb{E}_{s \sim d^\pi, a \sim \pi_\theta}\left[\nabla_\theta \log \pi_\theta(a|s) A^\pi(s, a)\right]$$

For a stochastic gradient ascent (SGA) update based on a single sample $(s_t, a_t)$ with learning rate $\alpha$, the parameter vector $\theta$ is updated in the direction of the gradient estimate:

$$\theta^{k+1} = \theta^k + \alpha \nabla_\theta \log \pi_\theta^k(a_t|s_t) A^k(s_t, a_t)$$

The change for a specific parameter $\theta_{s_t, a'}$, denoted by $\Delta\theta_{s_t, a'}$, is the component of this update vector corresponding to that parameter:

$$\Delta\theta_{s_t, a'} = \alpha \frac{\partial}{\partial \theta_{s_t, a'}} \log \pi_\theta^k(a_t|s_t) \cdot A^k(s_t, a_t)$$

For any state $s$ and action $a$, the log-policy is:

$$\log \pi_\theta(a|s) = \log\left(\frac{\exp(\theta_{s,a})}{\sum_{b \in \mathcal{A}(s)} \exp(\theta_{s,b})}\right)$$

$$= \theta_{s,a} - \log\left(\sum_{b \in \mathcal{A}(s)} \exp(\theta_{s,b})\right)$$

Differentiating with respect to a parameter $\theta_{s,a'}$ yields:

$$\frac{\partial}{\partial \theta_{s,a'}} \log \pi_\theta(a|s) = \frac{\partial \theta_{s,a}}{\partial \theta_{s,a'}} - \frac{\frac{\partial}{\partial \theta_{s,a'}} \sum_{b \in \mathcal{A}(s)} \exp(\theta_{s,b})}{\sum_{b \in \mathcal{A}(s)} \exp(\theta_{s,b})}$$

$$= \mathbb{I}(a = a') - \frac{\exp(\theta_{s,a'})}{\sum_{b \in \mathcal{A}(s)} \exp(\theta_{s,b})}$$

$$= \mathbb{I}(a = a') - \pi_\theta(a'|s)$$

where $\mathbb{I}(\cdot)$ is the indicator function.

Substituting this result back into the expression for $\Delta\theta_{s_t, a'}$, with $s = s_t$ and $a = a_t$, we get:

$$\Delta\theta_{s_t, a'} = \alpha\left(\mathbb{I}(a_t = a') - \pi_\theta^k(a'|s_t)\right) A^k(s_t, a_t)$$

This single expression can be split into the two cases:

- If $a' = a_t$, the indicator $\mathbb{I}(a_t = a_t) = 1$, so $\Delta\theta_{s_t, a_t} = \alpha\left(1 - \pi_\theta^k(a_t|s_t)\right) A^k(s_t, a_t)$.

- If $a' \neq a_t$, the indicator $\mathbb{I}(a_t = a') = 0$, so $\Delta\theta_{s_t, a'} = \alpha\left(0 - \pi_\theta^k(a'|s_t)\right) A^k(s_t, a_t) = -\alpha\pi_\theta^k(a'|s_t) A^k(s_t, a_t)$.

$\square$

# E  BENCHMARKS

## E.1  DATASETS AND METRICS

- **AirQA-Real** (Cao et al., 2025) is a dataset for evaluating question-answering capabilities over full-length academic documents. It consists of 553 challenging question-answer pairs, all of which are manually annotated by 16 AI researchers based on a corpus of 6,797 recent scientific papers. The questions in the AirQA-Real dataset cover a wide variety of task types such as single-document detail extraction, multi-document analysis, and paper retrieval. Due to its complexity, which spans categories like text, tables, images, and formulas, it has become a valuable benchmark for evaluating the performance of advanced RAG systems, and is widely used to test an agent's ability to handle realistic research scenarios.

- **SciDQA** (Singh et al., 2024) is a dataset for evaluating the deep reading comprehension ability of models on scientific literature. It consists of 2,937 challenging question-answer pairs. All of them are naturally derived from the peer-review discussions on the OpenReview platform, featuring questions from expert reviewers and answers from paper authors. The questions in the SciDQA dataset cover a wide variety of information sources such as figures, tables, equations, appendices, and even require reasoning across multiple documents. Due to the high quality and depth of its questions, it has become a popular benchmark for evaluating complex scientific text understanding, and is widely used to facilitate research beyond surface-level comprehension. For our evaluation, we select a subset of 775 instances and reformatted them to align with the data structure of the AirQA-Real benchmark.

For AirQA-Real, we implement instance-specific, execution-based evaluation by designing 18 functions with optional parameters, also utilized for reward router. To evaluate performance on SciDQA, we utilize an LLM-as-a-judge to score agent responses against reference answers. The scoring is conducted on a 0-10 scale in 0.5-point increments and is then normalized to a 0-100 scale for reporting in Table 1.

## E.2  CATEGORIES

Questions within the benchmarks are categorized based on their primary data modality into five types: **text**, **table**, **image**, **formula**, and **metadata**. Following table presents illustrative examples from each category.

Table 5: Examples from datasets.

| Category | Question | Answer Format |
|---|---|---|
| text | What are the main components of ERRA model? | Your answer should be a python list of strings, every element of the list is the name of the component directly mentionned in this paper. |
| table | On which language does LLaMA-2 13B with no removal reaches its second highest perplexity? | Your answer should be a word DIRECTLY FROM THE PDF WITHOUT ANY EXPLANATION. |
| image | In Figure 1, the presence of what operation divides the discretization process of continuous speech into two categories? | Your answer should be a python string. |
| formula | According to the methods proposed by this paper,how to calculate the bias Scores when aggregating attributions for tokens, instances and instructions respectively? | Your answer should be a python list of three elements, every element is a formula string in latex format. |
| metadata | Who is the corresponding author of this paper? | Your answer should be a python string about the name of the corresponding author. |

## F  CASE STUDY

---

**Case Trajectory in AirQA-Real (PaperCompass-7B)**

**[Question]:** What research advances are incorporated into the generative language model that used to generate associations in different languages in the anchor_pdf?
**[Answer Format]:** Your answer should be a python list of several strings.
**[Anchor PDF]:** 'fd81f90f-555d-5e99-835b-153c2cdb7303'
**[Reference Answer (Score Points)]:**[a] ["Compute-optimal scaling", "Improved dataset mixtures", "Architectural and objective improvements"]

**[Draft]:**
1. You should first use the 'ClassicRetrieve' tool to search for relevant context about research advances incorporated into the generative language model that generates associations in different languages.
2. Next, you should analyze the search results to identify the most relevant 'pdf_id' and use the 'RetrieveFromDatabase' tool to get the title and abstract of the paper from the database.
3. If the retrieved title and abstract do not directly answer the question, you should refine your search by using the 'ClassicRetrieve' tool again with a more specific query focusing on the generative language model and research advances.
4. Finally, you should use the 'GenerateAnswer' tool to provide the final answer based on the most relevant information you have identified.

---

**Interaction Turn 1**

**[Thought]:**
To answer the question, I will first perform a similarity search in the vectorstore to find relevant context about research advances incorporated into the generative language model that generates associations in different languages. I will use the "text_sentence_transformers_all_minilm_l6_v2" collection for this purpose.
**[Action]:**
ClassicRetrieve(query='What research advances are incorporated into the generative language model that used to generate associations in different languages?', limit=5)
**[Observation]:**
{"score":0.7122,"pdf_id":"fd81f90f-555d-5e99-835b-153c2cdb7303","page_number":2, "table_name":"chunks","column_name":"text_content","primary_key":"acafbdad-5c..."}
{"score":0.6914,"pdf_id":"a6ef6048-ad5a-5bec-bd71-050211a1a772","page_number":11, "table_name":"chunks","column_name":"text_content","primary_key":"1e1a8633-2c..."}
{"score":0.6858,"pdf_id":"690324ff-819c-50e1-9e0e-212ff0e4288a","page_number":13, "table_name":"chunks","column_name":"text_content","primary_key":"647610d9-aa..."}
{"score":0.6796,"pdf_id":"f92054f4-7190-53ad-9861-3cb1dc663ec0","page_number":7, "table_name":"chunks","column_name":"text_content","primary_key":"410764aa-fb..."}
{"score":0.6661,"pdf_id":"a44a8437-adfe-5b69-90a7-7f7b3d88050d","page_number":1, "table_name":"chunks","column_name":"text_content","primary_key":"0b877a14-6c..."}

**Interaction Turn 2**

**[Thought]:**
The most relevant result is the first one, which mentions PaLM-2 (Anil et al., 2023) as a generative language model used for generating associations in different languages. I will retrieve the title and abstract of the paper with the corresponding 'pdf_id' from the database to get more context.
**[Action]:**
RetrieveFromDatabase(sql="SELECT title, abstract FROM metadata WHERE paper_id = 'fd81f90f-555d-5e99-835b-153c2cdb7303';")
**[Observation]:**

---

{"title":"SeeGULL Multilingual: a Dataset of Geo-Culturally Situated Stereotypes","abstract":"While generative multilingual models are rapidly being deployed, their safety and fairness evaluations are largely limited to resources collected in English. This is especially problematic for evaluations targeting inherently socio-cultural phenomena such as stereotyping, where it is important to build multilingual resources that reflect the stereotypes prevalent in respective language communities. However, gathering these resources, at scale, in varied languages and regions pose a significant challenge as it requires broad socio-cultural knowledge and can also be prohibitively expensive. To overcome this critical gap, we employ a recently introduced approach that couples LLM generations for scale with culturally situated validations for reliability, and build SeeGULL Multilingual, a global-scale multilingual dataset of social stereotypes, containing over 25K stereotypes, spanning 23 pairs of languages and regions they are common in, with human annotations, and demonstrate its utility in identifying gaps in model evaluations."}

### Interaction Turn 3

[Thought]:
The retrieved title and abstract do not directly answer the question about research advances incorporated into the generative language model. I will refine my search by using a more specific query focusing on the generative language model and research advances.
[Action]:
ClassicRetrieve(query='What research advances are incorporated into the generative language model PaLM-2 for generating associations in different languages?', limit=5)
[Observation]:
{"score":0.7419,"pdf_id":"fd81f90f-555d-5e99-835b-153c2cdb7303","page_number":2,
"table_name":"chunks","column_name":"text_content","primary_key":"acafbdad-5c..."}
{"score":0.7357,"pdf_id":"82faa706-bd3f-5fad-9180-b96a624b3dfb","page_number":10,
"table_name":"chunks","column_name":"text_content","primary_key":"4df57063-0d..."}
{"score":0.7095,"pdf_id":"1e6eeeab-ba5c-508e-a693-62a9b39f2d92","page_number":1,
"table_name":"chunks","column_name":"text_content","primary_key":"a3b70ca0-c5..."}
{"score":0.6903,"pdf_id":"eb787b77-5188-5411-b0f8-406356623bac","page_number":3,
"table_name":"chunks","column_name":"text_content","primary_key":"004cf773-bc..."}
{"score":0.6810,"pdf_id":"f175c474-44cb-532e-bba1-d2316790de1c","page_number":7,
"table_name":"chunks","column_name":"text_content","primary_key":"62c2c2a7-44..."}

### Interaction Turn 4

[Thought]:
The most relevant result is the fourth one, which mentions PaLM-2 incorporating a diverse set of research advances. I will retrieve the title and abstract of the paper with the corresponding 'pdf_id' from the database to get more context.
[Action]:
RetrieveFromDatabase(sql="SELECT title, abstract FROM metadata WHERE paper_id = 'eb787b77-5188-5411-b0f8-406356623bac';")
[Observation]:
{"title":"PaLM 2 Technical Report","abstract":"We introduce PaLM 2, a new state-of-the-art language model that has better multilingual and reasoning capabilities and is more compute-efficient than its predecessor PaLM. PaLM 2 is a Transformer-based model trained using a mixture of objectives. Through extensive evaluations on English and multilingual language, and reasoning tasks, we demonstrate that PaLM 2 has significantly improved quality on downstream tasks across different model sizes, while simultaneously exhibiting faster and more efficient inference compared to PaLM. This improved efficiency enables broader deployment while also allowing the model to respond faster, for a more natural pace of interaction. PaLM 2 demonstrates robust reasoning capabilities exemplified by large improvements over PaLM on BIG-Bench and other reasoning tasks. PaLM 2 exhibits stable performance on a suite of responsible AI evaluations, and enables inference-time control over toxicity without additional overhead or impact on other capabilities. Overall, PaLM 2 achieves state-

of-the-art performance across a diverse set of tasks and capabilities. When discussing the PaLM 2 family, it is important to distinguish between pre-trained models (of various sizes), fine-tuned variants of these models, and the user-facing products that use these models. In particular, user-facing products typically include additional pre- and post-processing steps. Additionally, the underlying models may evolve over time. Therefore, one should not expect the performance of user-facing products to exactly match the results reported in this report."}

**Interaction Turn 5**

[Thought]:
The retrieved title and abstract provide a comprehensive overview of the research advances incorporated into PaLM-2. The abstract mentions that PaLM 2 has better multilingual and reasoning capabilities, is more compute-efficient, and demonstrates robust reasoning capabilities and stable performance on responsible AI evaluations. I will use this information to generate the answer.
[Action]:
GenerateAnswer(answer=['PaLM-2 has better multilingual and reasoning capabilities, is more compute-efficient, and demonstrates robust reasoning capabilities and stable performance on responsible AI evaluations.', 'PaLM 2 has significantly improved quality on downstream tasks across different model sizes while exhibiting faster and more efficient inference compared to PaLM.', 'PaLM 2 exhibits stable performance on a suite of responsible AI evaluations, and enables inference-time control over toxicity without additional overhead or impact on other capabilities.'])
[Answer]:
['PaLM-2 has better multilingual and reasoning capabilities, is more compute-efficient, and demonstrates robust reasoning capabilities and stable performance on responsible AI evaluations.', 'PaLM 2 has significantly improved quality on downstream tasks across different model sizes while exhibiting faster and more efficient inference compared to PaLM.', 'PaLM 2 exhibits stable performance on a suite of responsible AI evaluations, and enables inference-time control over toxicity without additional overhead or impact on other capabilities.'] ✓

---
[a]Unseen for agent.

Case Trajectory in AirQA-Real (Qwen2.5-14B-Instruct)

[Question]: What research advances are incorporated into the generative language model that used to generate associations in different languages in the anchor_pdf?
[Answer Format]: Your answer should be a python list of several strings.
[Anchor PDF]: 'fd81f90f-555d-5e99-835b-153c2cdb7303'
[Reference Answer (Score Points)]: ["Compute-optimal scaling", "Improved dataset mixtures", "Architectural and objective improvements"]

**Interaction Turn 1**

[Thought]:
I need to find the research advances incorporated into the generative language model that used to generate associations in different languages from the anchor_pdf. I will use the metadata and sections tables in the DuckDB database to retrieve this information.
[Action]:
RetrieveFromDatabase(sql="SELECT sections.section_title, sections.section_content FROM sections JOIN metadata ON sections.ref_paper_id = metadata.paper_id WHERE metadata.paper_id = 'fd81f90f-555d-5e99-835b-153c2cdb7303' AND sections.section_title LIKE '%language model%' AND sections.section_content LIKE '%different languages%';")

**[Observation]:**
[Warning]: The SQL execution result is empty, please check the SQL first.

**Interaction Turn 2**

**[Thought]:**
The SQL query did not return any results, which means there are no sections directly titled or content containing phrases related to "language model" and "different languages". I will try a more general approach by searching for sections that discuss generative language models and multilingual associations.

**[Action]:**
ClassicRetrieve(query='What research advances are incorporated into the generative language model that used to generate associations in different languages?', limit=5)

**[Observation]:**
{"score":0.7122,"pdf_id":"fd81f90f-555d-5e99-835b-153c2cdb7303","page_number":2, "table_name":"chunks","column_name":"text_content","primary_key":"acafbdad-5c...}
{"score":0.6914,"pdf_id":"a6ef6048-ad5a-5bec-bd71-050211a1a772","page_number":11, "table_name":"chunks","column_name":"text_content","primary_key":"1e1a8633-2c...}
{"score":0.6858,"pdf_id":"690324ff-819c-50e1-9e0e-212ff0e4288a","page_number":13, "table_name":"chunks","column_name":"text_content","primary_key":"647610d9-aa...}
{"score":0.6796,"pdf_id":"f92054f4-7190-53ad-9861-3cb1dc663ec0","page_number":7, "table_name":"chunks","column_name":"text_content","primary_key":"410764aa-fb...}
{"score":0.6661,"pdf_id":"a44a8437-adfe-5b69-90a7-7f7b3d88050d","page_number":1, "table_name":"chunks","column_name":"text_content","primary_key":"0b877a14-6c...}

**Interaction Turn 3**

**[Thought]:**
From the retrieved results, the first entry mentions that the generative language model used to generate associations in different languages is PaLM-2, which is trained on large multilingual text across hundreds of languages. This information is sufficient to answer the question.

**[Action]:**
GenerateAnswer(answer=['PaLM-2 is a generative language model trained on large multilingual text across hundreds of languages.'])

**[Answer]:**
['PaLM-2 is a generative language model trained on large multilingual text across hundreds of languages.'] ✗

---

Case Trajectory in SciDQA (PaperCompass-7B)

**[Question]:** What are the results of CyCLIP on ImageNet-A/R/V2/Sketch datasets?
**[Answer Format]:** Your answer should be concise free-form text string, directly answering the question(s).
**[Anchor PDF]:** '6a0a1c4d-3b62-56cb-a6e9-48b790274828'
**[Reference Answer]:** CyCLIP outperforms CLIP on all these datasets on zero-shot evaluation.

**[Draft]:**
1. You should first use the 'ClassicRetrieve' tool to search for the results of CyCLIP on ImageNet-A/R/V2/Sketch datasets by querying the vectorstore with the relevant text.
2. Next, you should analyze the search results to identify the most relevant entries that contain the results of CyCLIP on the specified datasets.
3. If the initial search does not yield the specific results, you should refine the query to focus more specifically on the datasets and the performance metrics of CyCLIP.
4. If the refined query still does not yield the specific results, you should use the 'RetrieveFromDatabase' tool to search for the results of CyCLIP on ImageNet-A/R/V2/Sketch

datasets in the database, focusing on the relevant sections or tables.

5. Once you have identified the relevant results, you should use the 'GenerateAnswer' tool to provide the final answer based on the identified results.

---

**Interaction Turn 1**

[Thought]:
To answer the question, I will first use the ClassicRetrieve action to search for the results of CyCLIP on ImageNet-A/R/V2/Sketch datasets by querying the vectorstore with the relevant text. I will use the collection "text_sentence_transformers_all_minilm_l6_v2" for this purpose, as it is a good general-purpose text embedding model.
[Action]:
ClassicRetrieve(query='results of CyCLIP on ImageNet-A/R/V2/Sketch datasets', limit=5)
[Observation]:
{"score":0.6317,"pdf_id":"6a0a1c4d-3b62-56cb-a6e9-48b790274828","page_number":5, "table_name":"chunks","column_name":"text_content","primary_key":"6f1f2264-97..."}
{"score":0.6300,"pdf_id":"6a0a1c4d-3b62-56cb-a6e9-48b790274828","page_number":5, "table_name":"chunks","column_name":"text_content","primary_key":"84631a60-03..."}
{"score":0.6166,"pdf_id":"6a0a1c4d-3b62-56cb-a6e9-48b790274828","page_number":2, "table_name":"chunks","column_name":"text_content","primary_key":"9961b917-7b..."}
{"score":0.6128,"pdf_id":"6a0a1c4d-3b62-56cb-a6e9-48b790274828","page_number":6, "table_name":"chunks","column_name":"text_content","primary_key":"f9e3a833-53..."}
{"score":0.6063,"pdf_id":"6a0a1c4d-3b62-56cb-a6e9-48b790274828","page_number":10, "table_name":"chunks","column_name":"text_content","primary_key":"00448cf5-37..."}

---

**Interaction Turn 2**

[Thought]:
The retrieved results provide some context about the performance of CyCLIP on the ImageNet-A, ImageNet-R, ImageNetV2, and ImageNetSketch datasets. However, the exact results for ImageNet-A/R/V2/Sketch datasets are not explicitly mentioned. I will now refine the query to focus more specifically on the datasets and the performance metrics of CyCLIP.
[Action]:
ClassicRetrieve(query='performance of CyCLIP on ImageNet-A R V2 Sketch datasets', limit=5)
[Observation]:
{"score":0.6420,"pdf_id":"6a0a1c4d-3b62-56cb-a6e9-48b790274828","page_number":5, "table_name":"chunks","column_name":"text_content","primary_key":"6f1f2264-97..."}
{"score":0.6386,"pdf_id":"6a0a1c4d-3b62-56cb-a6e9-48b790274828","page_number":2, "table_name":"chunks","column_name":"text_content","primary_key":"9961b917-7b..."}
{"score":0.6083,"pdf_id":"6a0a1c4d-3b62-56cb-a6e9-48b790274828","page_number":5, "table_name":"chunks","column_name":"text_content","primary_key":"84631a60-03..."}
{"score":0.6009,"pdf_id":"6a0a1c4d-3b62-56cb-a6e9-48b790274828","page_number":10, "table_name":"chunks","column_name":"text_content","primary_key":"00448cf5-37..."}
{"score":0.5993,"pdf_id":"6a0a1c4d-3b62-56cb-a6e9-48b790274828","page_number":6, "table_name":"chunks","column_name":"text_content","primary_key":"f9e3a833-53..."}

---

**Interaction Turn 3**

[Thought]:
The refined query has provided more specific information about the performance of CyCLIP on the ImageNet-A, ImageNet-R, ImageNetV2, and ImageNetSketch datasets. However, the exact results for ImageNet-A/R/V2/Sketch are not explicitly mentioned. I will now use the RetrieveFromDatabase action to search for the specific results in the database.
[Action]:

RetrieveFromDatabase(sql="SELECT text_content FROM chunks WHERE ref_paper_id = '6a0a1c4d-3b62-56cb-a6e9-48b790274828' AND text_content LIKE '%CYCLIP%' AND text_content LIKE '%ImageNet%' AND text_content LIKE '%A%' AND text_content LIKE '%R%' AND text_content LIKE '%V2%' AND text_content LIKE '%Sketch%';")

**[Observation]:**
{"text_content":"generated using attacks on trained ImageNet models. In Table 2, we evaluate the zero-shot classification accuracy of CYCLIP on four natural distribution shift benchmarks for the ImageNet dataset: ImageNetV2 [48], ImageNetSketch [57], ImageNet-A [27], and ImageNet-R [25]. For most of the distribution shift benchmarks, both CLIP and CYCLIP undergo a significant reduction in their zero-shot performance compared to the original ImageNet1K dataset (last three columns in"}

{"text_content":"Table 2: Zeroshot Classification on Natural Distribution Shifts (%) ImageNetV2 ImageNetSketch ImageNet-A ImageNet-R Top1 Top3 Top5 Top1 Top3 Top5 Top1 Top3 Top5 Top1 Top3 Top5 CLIP 16.91 29.28 34.99 10.37 19.15 24.20 4.23 11.35 16.88 24.32 39.69 47.20 CYCLIP 19.22 32.29 38.41 12.26 22.56 28.17 5.35 13.53 19.51 26.79 42.31 50.03 %GAIN +13.7 +10.3 +9.8 +18.2 +17.8 +16.4 +26.5 +19.2 +15.6 +10.2 +6.6 +6.0 this experiment by a significant margin of improvement (10 - 27%). This result indicates that having"}

| **Interaction Turn 4** |
| --- |

**[Thought]:**
The retrieved results provide the exact performance metrics of CyCLIP on the ImageNet-A, ImageNet-R, ImageNetV2, and ImageNetSketch datasets. The results indicate that CYCLIP outperforms CLIP with an average relative gain of +17% on ImageNet natural distribution shift benchmarks. Based on this information, I can now generate the final answer.

**[Action]:**
GenerateAnswer(answer='CYCLIP outperforms CLIP with an average relative gain of +17% on ImageNet natural distribution shift benchmarks.')

**[Answer]:**
CYCLIP outperforms CLIP with an average relative gain of +17% on ImageNet natural distribution shift benchmarks.    Score[a]: **8.5**

---

[a]Evaluated by gpt-4o-mini-2024-07-18

| Case Trajectory in SciDQA (Qwen2.5-14B-Instruct) |
| --- |

**[Question]:** What are the results of CyCLIP on ImageNet-A/R/V2/Sketch datasets?
**[Answer Format]:** Your answer should be concise free-form text string, directly answering the question(s).
**[Anchor PDF]:** '6a0a1c4d-3b62-56cb-a6e9-48b790274828'
**[Reference Answer]:** CyCLIP outperforms CLIP on all these datasets on zero-shot evaluation.

| **Interaction Turn 1** |
| --- |

**[Thought]:**
To answer the question, I need to find the results of CyCLIP on the ImageNet-A/R/V2/Sketch datasets. I will first try to search the metadata table for any mentions of these datasets and CyCLIP. If no relevant information is found, I will then try to search the vectorstore using the text modalities.

**[Action]:**
ClassicRetrieve(query='CyCLIP and ImageNet-A/R/V2/Sketch datasets', limit=5)

**[Observation]:**
{"score":0.5952,"pdf_id":"65d774b3-6bb6-5475-926a-0dfa3a2e9100","page_number":4,"table_name":"chunks","column_name":"text_content","primary_key":"1c8430f1-e3..."}

{"score":0.5878,"pdf_id":"6a0a1c4d-3b62-56cb-a6e9-48b790274828","page_number":2,
"table_name":"chunks","column_name":"text_content","primary_key":"9961b917-7b...}
{"score":0.5790,"pdf_id":"a017fa2d-2f53-5616-b5a4-2aca563cc758","page_number":5,
"table_name":"chunks","column_name":"text_content","primary_key":"011300f4-e0...}
{"score":0.5608,"pdf_id":"6a0a1c4d-3b62-56cb-a6e9-48b790274828","page_number":5,
"table_name":"chunks","column_name":"text_content","primary_key":"2d851c74-f4...}
{"score":0.5579,"pdf_id":"6a0a1c4d-3b62-56cb-a6e9-48b790274828","page_number":6,
"table_name":"chunks","column_name":"text_content","primary_key":"f9e3a833-53...}

**Interaction Turn 2**

**[Thought]:**
From the search results, I found that CyCLIP has been evaluated on ImageNet-A and ImageNet-V2 datasets. However, there is no direct mention of ImageNet-R and ImageNet-Sketch datasets. I will now try to search the vectorstore using the image modality to find any relevant images or regions that might contain information about these datasets.
**[Action]:**
ClassicRetrieve(query='CyCLIP and ImageNet-A/R/V2/Sketch datasets', limit=5, modality='image')
**[Observation]:**
[Error]: Failed to parse the parameters for action ClassicRetrieve from the response. ClassicRetrieve.__init__() got an unexpected keyword argument 'modality'.

**Interaction Turn 3**

**[Thought]:**
It seems that the 'modality' parameter is not supported in the current implementation of the ClassicRetrieve action. I will proceed with the text-based search and manually check if there is any mention of ImageNet-R and ImageNet-Sketch datasets in the retrieved results.
**[Action]:**
GenerateAnswer(answer='CyCLIP has been evaluated on ImageNet-A and ImageNet-V2 datasets. However, there is no direct mention of ImageNet-R and ImageNet-Sketch datasets in the retrieved results.')
**[Answer]:**
CyCLIP has been evaluated on ImageNet-A and ImageNet-V2 datasets. However, there is no direct mention of ImageNet-R and ImageNet-Sketch datasets in the retrieved results.
Score[a]: **1.5**

[a]Evaluated by gpt-4o-mini-2024-07-18

