# OpenReview forum: "Decomposing Scientific Paper Queries with Draft-and-Follow Policy Optimization to Narrow Knowing-Doing Gap"
_ICLR.cc/2026/Conference — ICLR 2026 Conference Desk Rejected Submission_

### Official Review · Reviewer_UDhF · 2025-10-31

**Soundness:** 3
**Presentation:** 3
**Contribution:** 3
**Rating:** 6
**Confidence:** 2

**Summary:**

This paper introduces a framework for improving scientific paper question answering (Paper-QA) by bridging the “knowing-doing gap” in large language models (LLMs). Although LLMs may “know” the correct reasoning steps, they often fail to execute them effectively. To address this, the authors propose a Draft-and-Follow Policy Optimization (DFPO) approach that separates high-level planning (“draft”) from concrete actions (“follow”).
PaperCompass integrates hierarchical reinforcement learning, optimizing both knowledge-level and action-level reasoning. It also introduces techniques such as negative sample masking and reward routing to stabilize training. Experiments on two benchmarks (AirQA-Real and SciDQA) show that DFPO significantly improves efficiency, achieving similar accuracy with fewer tool calls and enabling a 3B model to match the performance of much larger models.

**Strengths:**

-  The paper clearly defines the “knowing–doing gap” and provides a structured way to address it. The draft-and-follow approach neatly separates planning and execution, inspired by hierarchical reinforcement learning.
-  DFPO boosts efficiency without losing accuracy and allows smaller models to perform competitively on research-level QA tasks. The use of negative sample masking and reward routing improves training stability and prevents wasted updates.
-  The approach shows that small, structured models can handle complex reasoning tasks efficiently, which is valuable for research applications.

**Weaknesses:**

-  The experiments are limited to paper-QA tasks, so it’s uncertain whether the method generalizes to other reasoning or decision-making domains.
-  The multi-stage pipeline (fine-tuning + RL optimization) adds overhead and could be difficult for others to reproduce.
-  The framework relies on reliable tool-use environments, so its benefits may shrink in open-ended or less-structured settings.

**Questions:**

- The proposed DFPO framework is evaluated mainly on scientific paper QA tasks. Could the authors discuss how this method might generalize to other domains that also involve reasoning and action?
- The multi-stage training pipeline (DTFT + DFPO) seems computationally expensive. Can the authors provide an estimate of training cost compared to standard RL fine-tuning methods?

---

> ### Author Response · Authors · 2025-11-21
> **Response to Reviewer UDhF (1/2)**
>
> We thank the reviewer for the constructive feedback and address the comments below. We have highlighted the revised and newly added clarifications in **yellow** in the revised manuscript.
>
> ## **Weakness 1: Experiments are limited to paper-QA tasks**
>
> To address your concern, we evaluated Qwen2.5-7B-Instruct w/ SFT and PaperCompass-7B on both HotpotQA [1] and MusiQue [2], using the Search-R1 [3] implementation as the testing framework. Both models are evaluated under the same Search-R1 multi-turn retrieval-QA protocol with identical tools, max turns, and retriever settings. For retrieval, we use the 2018 Wikipedia dump as the knowledge source and E5 as the retriever. We report Answer Exact Match (EM) for HotpotQA and MusiQue:
>
> | **Models** | **HotpotQA** | **MusiQue** |
> | --- | --- | --- |
> | Qwen2.5-7B-Instruct w/ SFT | 0.238 | 0.052 |
> | PaperCompass-7B | 0.291 | 0.133 |
>
> Notably, both Qwen2.5-7B-Instruct w/ SFT and PaperCompass-7B are checkpoints trained by us solely on the Paper-QA dataset, and are evaluated on HotpotQA and MusiQue to assess their generalization. We did not train these models on HotpotQA or MusiQue and then re-evaluate them. The results strictly reflect generalization rather than in-domain fine-tuning.
>
> - **The reason for taking Qwen2.5-7B-Instruct w/ SFT as the baseline.** We note that the Paper-QA training data used in our work differs substantially in distribution from HotpotQA and MusiQue. To ensure a fair comparison, we therefore take Qwen2.5-7B-Instruct fine-tuned on the Paper-QA pairs as the baseline, and compare it against PaperCompass-7B. This setup controls for all factors unrelated to DTFT and DFPO, ensuring that any observed performance differences arise solely from the proposed training framework. Our goal here is not to re-benchmark all Agentic-RAG methods on HotpotQA and MusiQue, but to isolate the DTFT and DFPO under identical model and data conditions.
> - **Performance analysis.** Although PaperCompass-7B performs worse on HotpotQA and MusiQue than in-domain fine-tuned models, it is crucial to emphasize that the Paper-QA training data differs substantially in distribution from both benchmarks. Therefore, to meaningfully evaluate generalization, the appropriate comparison should be made against Qwen2.5-7B-Instruct w/ SFT trained on the same Paper-QA data, rather than models fine-tuned on unrelated datasets. As shown in the table above, PaperCompass-7B achieves markedly better performance than Qwen2.5-7B-Instruct w/ SFT, demonstrating its superior ability to generalize beyond the domain of its training data.
>
> **References**:
>
> [1] HotpotQA: A Dataset for Diverse, Explainable Multi-hop Question Answering, EMNLP 2018.
>
> [2] MuSiQue: Multihop Questions via Single-hop Question Composition, TACL 2022.
>
> [3] Search-R1: Training LLMs to Reason and Leverage Search Engines with Reinforcement Learning, COLM 2025.
>
> ## **Weakness 2: Multi-stage pipeline adds overhead, making hard to reproduce**
>
> We thank the reviewer for raising this point. We address your concern from two perspectives: **pipeline complexity** and **computational cost**.
>
> 1. **Pipeline complexity.** PaperCompass introduces an additional draft-generation step before DTFT and a modified objective that separately computes draft and solution advantages in DFPO. While this adds a lightweight component to the standard SFT+GRPO workflow, it is consistent with many recent RL fine-tuning pipelines [1,2,3] that also modify SFT (e.g., via specialized demonstrations) or GRPO/PPO (e.g., reward shaping, trajectory augmentation). PaperCompass belongs to the same family of **SFT modification + GRPO modification** approaches and requires no new model components, no reward model, and no special infrastructure.
> 2. **Computational cost.** We further quantify training overhead by comparing total NPU-hours for DTFT+DFPO versus SFT+GRPO on Ascend 910B4 (**detailed in Rebuttal for Question 2**). DTFT+DFPO requires **~1.7×** the training cost of a standard SFT+GRPO pipeline, which is comparable to many multi-stage RLFT setups. Importantly, DFPO yields **20–30% efficiency gains in downstream multi-turn interactions**, allowing the extra computation to offset.
>
> To ensure reproducibility, we have open-sourced the code, improved documentation, and will release the **synthetic training data** in the camera-ready version. The entire pipeline can be reproduced with standard training scripts and requires only minor configuration changes relative to vanilla GRPO.
>
> **References**:
>
> [1] WebThinker: Empowering Large Reasoning Models with Deep Research Capability, NeurIPS 2025.
>
> [2] S-GRPO: Early Exit via Reinforcement Learning in Reasoning Models, NeurIPS 2025.
>
> [3] RAGEN: Understanding Self-Evolution in LLM Agents via Multi-Turn Reinforcement Learning, arXiv: 2504.20073.

---

> > ### Author Response · Authors · 2025-11-21
> > **Response to Reviewer UDhF (2/2)**
> >
> > ## **Weakness 3: Framework relies on reliable tool-use environments**
> >
> > To address your concern regarding unreliable tool environments, we introduced three kinds of tool unreliability to our environment for **RetrieveFromVectorstoreNoise** and **RetrieveFromDatabaseNoise**:
> >
> > - **Call-failure noise.** Each tool call fails with probability $p_f=0.1$, returning a function-call failure output.
> > - **Top-k corruption noise.** With probability $p_c=0.1$, we replace retrieved top-k items with irrelevant ones.
> > - **Structural noise.** Randomly return empty or malformed tool outputs.
> >
> > These settings collectively simulate real-world unreliability beyond simple random failures.  We followed the same training procedure described in our paper to train PaperCompass-7B-Noise under this unreliable tool-calling environment, and evaluated both PaperCompass-7B-Noise and Qwen2.5-7B-Instruct w/ SFT under the same noisy tool environment at test time.:
> >
> > **AirQA-Real**
> >
> > | **Models** | **Avg.** | **I-Avg.** | **Valid Answer** | **Interaction Turns** |
> > | --- | --- | --- | --- | --- |
> > | Qwen2.5-7B-Instruct w/ SFT | 20.5 | 14.3 | 67.2 | 10.3 |
> > | PaperCompass-7B-Noise | 23.6 | 18.7 | 75.0 | 7.38 |
> >
> > **SciDQA**
> >
> > | **Models** | **Avg.** | **I-Avg.** | **Valid Answer** | **Interaction Turns** |
> > | --- | --- | --- | --- | --- |
> > | Qwen2.5-7B-Instruct w/ SFT | 42.5 | 23.9 | 46.8 | 13.7 |
> > | PaperCompass-7B-Noise | 45.9 | 33.3 | 66.9 | 9.46 |
> >
> > Compared to the original setting, PaperCompass-7B-Noise shows a smaller performance drop than the SFT baseline (e.g., on AirQA-Real, −7.5% vs. −9.3% in Avg; -15.7% vs. -18.8% in I-Avg), indicating higher robustness.
> >
> > ## **Question 1: How this method might generalize to other domains?**
> >
> > We believe that the training pipeline of PaperCompass is generalizable to other domains that involve both reasoning and action.
> >
> > PaperCompass is designed for tasks that require both long-horizon reasoning and action, and our pipeline separates these two roles explicitly: the **draft** serves as coarse-grained planning, while the **solution** follows a ReAct-style execution that is grounded in tool feedback. This two-level structure provides a domain-agnostic inductive bias: planning focuses on goal abstraction, and execution focuses on environment alignment, which is a common requirement across many agentic tasks. Plan-and-Act [1] adopts a similar idea, but their feedback on planning is based on natural language, while we provide feedback directly through gradients.
> >
> > Our key contribution, **DFPO**, is an algorithm-level optimization that decomposes gradients for draft vs. solution, thereby reducing the knowing–doing gap without assuming any domain-specific features. The only assumption needed is the availability of a verifiable interaction signal over multi-turn trajectories, which holds for a wide range of tool-augmented agents.
> >
> > Empirically, we have already observed cross-domain generalization within QA: models trained only on Paper-QA transfer to HotpotQA and MuSiQue under the same Search-R1 interaction protocol and still outperform an SFT baseline, indicating that our training framework improves robustness beyond the scientific domain.
> >
> > We agree that validating the *entire pipeline* on non-QA domains (e.g., web navigation or code/tool agents) is a valuable next step. Given that these domains share the same planning–execution interaction pattern, we expect DFPO to remain applicable, and we plan to include a pilot study along this line in the extended version.
> >
> > **References**:
> >
> > [1] Plan-and-Act: Improving Planning of Agents for Long-Horizon Tasks, ICML 2025.
> >
> > ## **Question 2: An estimate of training cost compared to standard RL fine-tuning methods**
> >
> > To clarify the computational cost of our method, we provide a precise comparison of the total training budget of largest model in our training procedure measured in **NPU-hours** (number of NPUs × wall-clock time). All experiments were conducted on Ascend 910B4 NPUs. The breakdown is shown below:
> >
> > | **Pipeline** | **Stage** | **#Examples / Steps** | **NPUs** | **Training Time** | **Npu-Hours** |
> > | --- | --- | --- | --- | --- | --- |
> > | DTFT + DFPO | DTFT | 10,000 examples | 2 | 35.05 hours | 70.10 |
> > |  | DFPO | 400 steps | 8 | 22.32 hours | 178.53 |
> > |  | Total | - | - | - | 248.63 |
> > | SFT + GRPO | SFT | 4,000 examples | 2 | 7.57 hours | 15.13 |
> > |  | GRPO | 400 steps | 8 | 16.58 hours | 132.67 |
> > |  | Total | - | - | - | 147.80 |
> >
> > Our full **DTFT + DFPO pipeline** requires **248.63 NPU-hours**, while the baseline **SFT + GRPO** pipeline requires **147.80 NPU-hours**, meaning our method incurs roughly **1.7×** more compute under identical hardware and training-step configurations. We emphasize that this additional cost stems primarily from **optimizing both draft planning and solution reasoning**, which is central to improving long-horizon reasoning consistency and reducing the knowing–doing gap.

---

> > > ### Author Response · Authors · 2025-11-28
> > > **We are looking forward to further discussing with you**
> > >
> > > Dear Reviewer UDhF,
> > >
> > > I hope this message finds you well. As the discussion period is nearing its end with less than one week remaining, I wanted to ensure we have addressed all your concerns satisfactorily. If there are any additional points or feedback you would like us to consider, please let us know. Your insights are invaluable to us, and we are eager to address any remaining issues to improve our work.
> > >
> > > Thank you for your time and effort in reviewing our paper!
> > >
> > > Best regards,
> > >
> > > The authors of Submission 3227

---

### Official Review · Reviewer_cwPa · 2025-10-31

**Soundness:** 3
**Presentation:** 3
**Contribution:** 3
**Rating:** 6
**Confidence:** 2

**Summary:**

This paper introduces a framework called papercompass for extracting information from papers by agents. Specifically, it generates a draft first, then lets the agent work with it. To enhance this process, they also introduced an RL method called DFPO, which could enhance model performance for larger models on PaperQA.

**Strengths:**

1. This paper is well structured and comprehensive

2. This paper introduces a novel framework called Papercompass, a fine-tuning method called DTFT and a specific RL method called Draft-Follow Policy Optimization.

3. Papercompass can enhance the capabilities of small models, resulting in lower deployment costs compared to using larger models.

**Weaknesses:**

1. Most results in this paper are lacking human feedback. How does a researcher evaluate the performance of your model?

2. The abbreviation “DTFT” is introduced abruptly in the text, which makes the reading confusing.

**Questions:**

Good and comprehensible work. Can your method (e.g., Papercompass 7B) extract key information effectively from this paper based on your judgment? Compared to a result using the prompt generated from papercompass by a full model, which one do you think is better?

---

> ### Author Response · Authors · 2025-11-21
> **Response to Reviewer cwPa (1/2)**
>
> We thank the reviewer for the constructive feedback and address the comments below. We have highlighted the revised and newly added clarifications in **yellow** in the revised manuscript.
>
> ## **Weakness 1: Lacking human feedback**
>
> While the underlying benchmarks [1,2] are indeed human-annotated, our main reported metrics (Avg, I-Avg) are still automatic proxies derived from these labels. We agree with the reviewer that it is important to additionally evaluate model behavior from a human researcher’s perspective. To this end, we further conducted an expert evaluation of trajectories, generated from PaperCompass-7B and Qwen2.5-7B-Instruct on 20 randomly sampled questions from AirQA-Real and SciDQA. For each question, an experienced NLP researcher was shown two anonymized trajectories (PaperCompass-7B vs. Qwen2.5-7B-Instruct) in random order and asked to rate **retrieval quality**, **tool-use soundness**, and **overall usefulness** on 1–5 scales.
>
> | **Models** | **retrieval quality** |  **tool-use soundness** | **overall usefulness** |
> | --- | --- | --- | --- |
> | PaperCompass-7B | 4.6 | 4.2 | 3.6 |
> | Qwen2.5-7B-Instruct | 3.8 | 3.6 | 2.6 |
>
> Compared with Qwen2.5-7B-Instruct, PaperCompass-7B has obvious improvements in the following aspects:
>
> 1. **High retrieval accuracy**.  When performing single-document retrieval, PaperCompass-7B demonstrates a strong understanding of the task specification and achieves a high success rate in locating the correct document based on its unique identifier (`paper_id`). In contrast, Qwen2.5-7B-Instruct sometimes fails to retrieve the corresponding paper, primarily because it does not reliably interpret `paper_id`, even when this is explicitly stated in the user prompt.
>
> ```sql
> # From PaperCompass-7B
> [Action]:
> RetrieveFromDatabase(sql="SELECT page_content FROM pages JOIN metadata ON pages.ref_paper_id = metadata.paper_id WHERE metadata.paper_id = 'b72f144b-ccbe-500a-862f-dcfc1b4d0f61' AND pages.page_number = 5;")
> ```
>
> 2. **Clever tool-use strategy.** Comprehensive questions (requiring the system to first locate the target paper and subsequently identify specific content within it) do not explicitly provide the `paper_id` . PaperCompass consistently adopts a two-stage retrieval strategy: it first applies **ClassicRetrieve** for coarse filtering, followed by **RetrieveFromDatabase** for fine-grained lookup. In contrast, Qwen2.5-7B-Instruct occasionally invokes **RetrieveFromDatabase** directly. Without having obtained the correct `paper_id`, the model lacks the necessary grounding to access the target document and therefore tends to generate hallucinated content.
>
> ```sql
> # From Qwen2.5-7B-Instruct
> [Action]:
> RetrieveFromDatabase(sql="SELECT section_content FROM sections WHERE ref_paper_id = '837eaf5a-a9f7-55ba-99bb-7f470f0624cb' AND section_content LIKE '%Section 4%' AND section_content LIKE '%error bar%';")
> [Observation]:
> [Warning]: The SQL execution result is empty, please check the SQL first.
> ```
>
> This expert-review setup also provides a practical protocol for researchers to evaluate PaperCompass (or other agents) by sampling questions, inspecting trajectories, and rating retrieval quality, tool-use soundness, and usefulness.
>
> **References**:
>
> [1] NeuSym-RAG: Hybrid Neural Symbolic Retrieval with Multiview Structuring for PDF Question Answering, ACL 2025 Main.
>
> [2] SciDQA: A Deep Reading Comprehension Dataset over Scientific Papers, EMNLP 2024 Main.
>
> ## **Weakness 2: The abbreviation “DTFT”**
>
> Thank you for pointing this out. In the revised version, we have added the full name and a clear description of **DTFT (Draft-and-Follow Fine-Tuning)** at its first occurrence in the main text (Figure 1). We also ensured consistent terminology throughout the paper to avoid confusion.

---

> ### Author Response · Authors · 2025-11-21
> **Response to Reviewer cwPa (2/2)**
>
> ## **Question 1: Can PaperCompass-7B extract key information effectively from this paper?**
>
>  We clarify that **PaperCompass does not generate prompts for another full model**.  Instead, it is an autonomous retrieval-augmented agent that directly answers the question through a multi-step pipeline (retrieval, document identification, targeted reading, and answer generation). Based on this understanding, we answer your two question point by point:
>
> 1. **Can PaperCompass-7B extract key information effectively from this paper?** Based on the content of this paper, we formulated five evaluation questions for PaperCompass-7B, and its responses are provided below.
>
>
>     | **Question** | **Answer** |
>     | --- | --- |
>     | What is the main contribution of this paper？ | PaperCompass introduces a Draft-and-Follow agent framework plus a new RL algorithm DFPO to bridge the knowing–doing gap and make small LLM agents more efficient in Paper-QA. |
>     | What is the relationship between DFPO and M-GRPO? | DFPO is essentially M-GRPO plus a draft-bias term, making it jointly optimizes draft and solution. |
>     | Tell me the name of benchmarks are used in experiments. | The experiments use AirQA-Real and SciDQA. |
>     | Introduce the training pipeline of PaperCompass. | PaperCompass first performs Draft & Tool-Use Fine-Tuning on synthetic trajectories, then applies DFPO to jointly optimize draft generation and solution execution. |
>     | Which three metrics do the authors use to show efficiency of PaperCompass in Figure 4? | Figure 4 evaluates efficiency with Correct Rate, Valid Answers, and Interaction Turns. |
> 2. **What about using the prompt generated from PaperCompass by a full model?** According to PaperCompass’s pipeline, we interpret the “prompt” mentioned by the reviewer as referring to the draft.  Under this assumption, we argue that using a full model (e.g., Qwen2.5-72B-Instruct) to generate result based on draft within PaperCompass framework would further improve performance.  As shown in our rebuttal to your weakness 1, PaperCompass demonstrates clever tool-use strategy.  However, as emphasized in Section 4.5, deep semantic understanding remains a limitation of the 7B-scale model: even when the agent retrieves the correct evidence, it may still fail to fully comprehend it.  A full model, on the other hand, is capable of handling such cases effectively.  Nevertheless, our contribution is to enable the **same agent** to optimize both cognition and action simultaneously, which is precisely what we emphasize in the paper as addressing the “knowing–doing gap.” We do not aim to outperform much larger models.

---

> > ### Author Response · Authors · 2025-11-28
> > **We are looking forward to further discussing with you**
> >
> > Dear Reviewer cwPa,
> >
> > I hope this message finds you well. As the discussion period is nearing its end with less than one week remaining, I wanted to ensure we have addressed all your concerns satisfactorily. If there are any additional points or feedback you would like us to consider, please let us know. Your insights are invaluable to us, and we are eager to address any remaining issues to improve our work.
> >
> > Thank you for your time and effort in reviewing our paper!
> >
> > Best regards,
> >
> > The authors of Submission 3227

---

### Official Review · Reviewer_YEmS · 2025-10-31

**Soundness:** 2
**Presentation:** 1
**Contribution:** 3
**Rating:** 4
**Confidence:** 5

**Summary:**

This study addresses efficiency issues in existing LLM agents when processing scientific papers, particularly the tendency of small-scale models to fall into repetitive retrieval loops. The core innovation lies in introducing a "Draft-and-Follow" architecture inspired by cognitive science, which requires the agent to first generate a high-level planning draft before executing fine-grained tool calls based on that draft. To support this architecture, the researchers developed a specialized reinforcement learning algorithm called DFPO (Draft-and-Follow Policy Optimization), which can simultaneously optimize draft quality and final solutions. Experimental results demonstrate that PaperCompass based on Qwen2.5-3B achieves performance comparable to 32B parameter models on the AirQA-Real benchmark, while significantly improving the interaction efficiency metric I-Avg. Across two major benchmarks (AirQA-Real and SciDQA), the method reduces tool calls by 31.3% without sacrificing accuracy.

**Strengths:**

**1. Comprehensive Experimental Design**

The research includes extensive ablation studies and analyses, encompassing entropy dynamics analysis, repetition behavior analysis, and independent contribution evaluation of each component. The paper provides detailed efficiency statistics across multiple dimensions, including correct rate, valid answers, and interaction turns, offering thorough empirical validation of the proposed approach.

**2. Strong Reproducibility**

The paper provides complete experimental settings and implementation details. The authors have made their code publicly available and documented detailed hyperparameters, including maximum interaction turns, temperature settings, and training configurations. This level of transparency facilitates reproduction and verification of the results by the research community.

**3. Thoughtful innovation**

The paper cleverly applies the draft-solution paradigm and designs the DFPO algorithm to assign different advantages to draft and solution components. The framework represents a novel streamlined implementation of RL designed to bridge the 'knowing-doing' gap observed in LLMs. The integration of negative sample masking and reward router mechanisms demonstrates thoughtful engineering to address practical training challenges.

**Weaknesses:**

**1. Excessive Theoretical Derivations with Limited Substance**

The authors attempt to justify their method design through theoretical proofs, but most of these proofs merely restate algorithmic definitions without providing genuine theoretical contributions.

**(1) Misleading "Regularization" Claim:** The claim that DFPO's policy gradient is a "regularized form" of M-GRPO is imprecise and the derivation lacks theoretical contribution. The authors first define J_DFPO to contain both draft and solution advantage terms, then "prove" it can be written as M-GRPO plus an additional term. However, this additional term has no independent theoretical origin—it is purely a product of algebraic rearrangement. The derivation employs only trivial mathematical manipulations: (a) distributive property: decomposing Â^draft into Â^solution + (Â^draft - Â^solution), and (b) symbolic reorganization: defining ΔÂ = (Â^draft - Â^solution).

The "regularization term" in DFPO neither prevents overfitting nor represents an independently added constraint—it is simply a relabeled portion of the original objective function. The paper's use of the term "regularization" is highly misleading, as this decomposition is a direct consequence of the objective function's definition. This represents an engineering observation rather than a theoretical contribution.

**(2) Circular Reasoning in Negative Sample Masking:** The actual operation of negative sample masking is straightforward—when the solution is incorrect, directly set the draft advantage to the solution advantage (typically negative). However, the authors construct an elaborate mathematical framework through Theorem 2, Proposition 1, and Lemma 3 to "prove" the rationality of this heuristic rule, which essentially constitutes tautological reasoning.

The logical structure forms a circle: **Premise A**: Negative sample masking sets draft advantage to negative values for wrong solutions; **Theorem 2**: Proves this setting prevents erroneous reinforcement; **Proof method**: Assumes negative sample masking is already in effect, then demonstrates advantages are negative; **Conclusion**: Negative sample masking is justified (because Theorem 2 says it is justified).

This is a case of "using rules to prove rules"—circular argumentation. The paper does not derive the necessity of negative sample masking from independent theoretical foundations; instead, it first defines this rule, then "proves" it conforms to its own definition. The core contribution can be stated in one sentence: "When the solution is wrong, we do not update the draft policy to avoid reinforcing erroneous planning paths". This is far clearer than the current Theorem 2-Lemma 2-Proposition 1 system.

I do not believe that top-tier ML conferences necessarily require theoretical derivations. The authors seem to feel that simple, effective methods are "insufficiently academic," and thus compensate through mathematical packaging—but this approach undermines the paper's credibility and readability.

**2. Marginal and Sometimes Negative Performance Gains**

On the AirQA-Real benchmark, PaperCompass demonstrates quite limited accuracy improvements. Specifically, for the Qwen2.5-3B model-based version, accuracy only increased from 22.2% (SFT baseline) to 23.7% (final), an improvement of merely 1.5 percentage points. For the 7B model, the situation is slightly better but still not significant: from 23.9% (SFT) to 25.3% (DFPO), an increase of approximately 1.4 percentage points. In certain subcategories, the 7B model even exhibits negative growth. These results warrant further evaluation of whether the method is genuinely effective.

**3. Lack of Statistical Significance Validation**

The paper lacks multiple repeated experiments and significance testing. Given that some improvements are marginal (as pointed out in Weakness 2), without significance testing it is difficult to determine whether these represent genuine improvements or merely noise in the experimental results.

**Questions:**

**Regarding Draft Length and Training Signal Strength:**

What is the average token length of the draft in the current implementation? What are the average token  lengths of think and answer components per turn? If the draft length constitutes a low proportion across multi-turn interactions, it may be drowned out in batch loss computation. Can the draft be sufficiently learned under such conditions? This concern is particularly relevant given that the draft is crucial to the proposed framework's effectiveness, and insufficient training signal could undermine the core contribution of the work.

---

> ### Author Response · Authors · 2025-11-21
> **Response to Reviewer YEmS (1/2)**
>
> We thank the reviewer for the constructive feedback and address the comments below. We have highlighted the revised and newly added clarifications in **yellow** in the revised manuscript.
>
> ## **Weakness 1.1: Misleading "Regularization" Claim**
>
> Thank you for raising this concern. We apologize for the confusion caused by our terminology. Our intention was not to claim that DFPO introduces an independently designed or theoretically novel regularizer. Although the derivation is algebraically simple, the decomposition is not arbitrary, it follows from two deliberate design choices in DFPO:
>
> 1. **Unified normalization of the draft and solution.** DFPO normalizes the combined (draft + solution) sequence rather than treating the two segments independently.  This coupling yields a single objective whose gradient naturally decomposes into the GRPO component plus an additional bias term.
> 2. **The draft is defined as a prefix of the full agent response.** This structural constraint enables the expectation factorization required for applying the distributive property, making the decomposition valid.
>
> The purpose of Theorem 1 is therefore **explanatory**: it clarifies why DFPO’s update differs from GRPO and how an implicit bias emerges from our design choices.  It is not intended as a new theoretical contribution or the introduction of an explicit regularizer. We have updated the wording accordingly in the revised version to avoid any impression of unnecessary mathematical formalism (Section 3.3).
>
> ## **Weakness 1.2: Circular Reasoning in Negative Sample Masking**
>
> We appreciate the reviewer’s detailed reading and apologize that the dependency structure among Theorem 2, Proposition 1, and Lemma 3 was not sufficiently clear. We would like to emphasize that the proof sequence does **not** rely on circular reasoning:
>
> 1. **Theorem 2 does not assume negative sample masking.**  It analyzes the advantage under the **unmasked** DFPO objective and shows that, for incorrect solutions, the draft advantage becomes negative due to the reward design.
> 2. **Negative sample masking is introduced after Theorem 2 as a technical choice.**  Motivated by Theorem 2’s observation, negative sample masking is added to prevent reinforcing erroneous planning trajectories in PaperCompass.
> 3. **Proposition 1 and Lemma 3 analyze the consequences given that masking is applied.**
> These results are intended to characterize the optimization dynamics (e.g., entropy and update direction) conditional on the masking rule being used, while not to prove the validity of the rule itself.  **This is an analysis of effects, not a justification of necessity.**
>
> All in all, our narrative regarding Negative Sample Masking proceeds as follows:
>
> **Theorem 2 (no masking)→ motivates** Negative Sample Masking (techinal choice) **→ then** Proposition 1 / Lemma 3 (effect analysis given masking)
>
> We have revised several statements in the revised version (Appendix D) to clarify that our objective is to analyze how Negative Sample Masking influences the parameter update direction and entropy dynamics. Our intention is **not** to justify the method’s rationality based solely on observations that arise after introducing Negative Sample Masking.
>
> ## **Weakness 2: Marginal and Sometimes Negative Performance Gains**
>
> We would like to clarify that the primary goal of PaperCompass is **to improve interaction efficiency (I-Avg)** while **preserving accuracy**, rather than maximizing accuracy itself.
>
> Across both datasets, DFPO achieves **slightly higher accuracy than SFT** (typically +0.5% to +1.4% depending on model size), while also delivering **substantial improvements in interaction efficiency** (e.g., I-Avg: 16.7 → 20.0 for 3B; 17.8 → 21.0 for 7B). Figure 4 in our paper further illustrates the magnitude of these efficiency gains. Minor variations in individual subcategories fall within the expected noise range for multi-turn agentic QA, particularly given the **category imbalance in AirQA-Real** (e.g., the “metadata” type contains only 22 questions) [1]. Such subcategory-level fluctuations reflect **micro-variance due to skewed sample sizes**, and do not contradict the overall trend: DFPO consistently improves both efficiency and overall accuracy across datasets and model scales.
>
> **References**:
>
> [1] NeuSym-RAG: Hybrid Neural–Symbolic Retrieval with Multiview Structuring for PDF Question Answering, ACL 2025 Main.

---

> > ### Author Response · Authors · 2025-11-21
> > **Response to Reviewer YEmS (2/2)**
> >
> > ## **Weakness 3: Lack of Statistical Significance Validation**
> >
> > Thank you for pointing out the importance of repeated experiments and statistical significance.
> > To address this concern, we re-ran all experiments **under 5 different random seeds** (42, 0, 1, 2, 3), for both 3B and 7B models, across **all core metrics** on AirQA-Real and SciDQA. We report **mean ± standard deviation** and conduct **paired t-tests** between DFPO and SFT. Across all settings and metrics, DFPO consistently improves over SFT, and all differences are statistically significant (p < 0.01, most p < 1e-4).
> >
> > **AirQA-Real (Qwen2.5-7B-Instruct w/ SFT vs. PaperCompass-7B)**
> >
> > | Metric | SFT | DFPO | p-value |
> > | --- | --- | --- | --- |
> > | **Avg** | 23.38 ± 0.47 | **25.30 ± 0.41** | 2.3e-4 |
> > | **I-Avg** | 17.24 ± 0.36 | **20.98 ± 0.29** | 1.4e-5 |
> > | **Valid Answers** | 69.78 ± 1.50 | **85.04 ± 1.16** | 7.3e-6 |
> > | **Interaction Turns (↓)** | 9.13 ± 0.15 | **6.24 ± 0.09** | 1.2e-6 |
> >
> > **SciDQA (Qwen2.5-7B-Instruct w/ SFT vs. PaperCompass-7B)**
> >
> > | Metric | SFT | DFPO | p-value |
> > | --- | --- | --- | --- |
> > | **Avg** | 45.20 ± 0.70 | **48.34 ± 0.48** | 1.9e-4 |
> > | **I-Avg** | 32.66 ± 0.43 | **37.46 ± 0.37** | 2.6e-5 |
> > | **Valid Answers** | 64.04 ± 1.23 | **80.94 ± 0.73** | 1.0e-5 |
> > | **Interaction Turns (↓)** | 9.62 ± 0.17 | **7.99 ± 0.04** | 2.7e-5 |
> >
> > **AirQA-Real (Qwen2.5-3B-Instruct w/ SFT vs. PaperCompass-3B)**
> >
> > | Metric | SFT | DFPO | p-value |
> > | --- | --- | --- | --- |
> > | **Avg** | 21.70 ± 0.52 | **23.76 ± 0.19** | 7.0e-4 |
> > | **I-Avg** | 16.36 ± 0.35 | **19.96 ± 0.09** | 1.1e-5 |
> > | **Valid Answers** | 71.92 ± 1.31 | **86.92 ± 0.79** | 4.1e-5 |
> > | **Interaction Turns (↓)** | 8.68 ± 0.16 | **5.94 ± 0.16** | 1.1e-5 |
> >
> > **SciDQA (Qwen2.5-3B-Instruct w/ SFT vs. PaperCompass-3B)**
> >
> > | Metric | SFT | DFPO | p-value |
> > | --- | --- | --- | --- |
> > | **Avg** | 42.86 ± 0.25 | **43.76 ± 0.18** | 8.6e-3 |
> > | **I-Avg** | 28.06 ± 0.19 | **36.18 ± 0.22** | 5.7e-7 |
> > | **Valid Answers** | 57.16 ± 0.86 | **84.36 ± 0.50** | 5.2e-7 |
> > | **Interaction Turns (↓)** | 11.36 ± 0.11 | **6.36 ± 0.13** | 8.4e-7 |
> >
> > ## **Questions 1: Regarding Draft Length and Training Signal Strength**
> >
> > Thank you for raising this important question. We provided the full draft and solution token statistics below (you can also refer to the case study in Appendix F to understand the situation of the draft and solution under the specific problem). Across both datasets and both model scales, the draft occupies a **substantial portion** of the full trajectory, ensuring that its training signal is not drowned out.
> >
> > **Draft & Solution Token Lengths (average per example)**
> >
> > | Model | Dataset | Draft | Solution |
> > | --- | --- | --- | --- |
> > | **PaperCompass-7B** | AirQA-Real | **127.4** | 397.1 |
> > |  | SciDQA | **115.8** | 523.9 |
> > | **PaperCompass-3B** | AirQA-Real | **106.9** | 358.7 |
> > |  | SciDQA | **104.7** | 470.3 |
> > 1. **Draft is not short relative to total response.**  Across all settings, the draft contributes **20–30%** of the full response length, which is large enough to receive clear gradient signal during training.
> > 2. **DFPO preserves a strong per-token draft signal.**  We design separate reward functions for the draft and the solution, while normalizing them ****jointly over the full response length in the DFPO objective (rather than normalizing draft and solution segments independently). This normalization in a single trajectory [1,2] ensures that each token in the draft and solution segments receives the same per-token training intensity. As a result, the draft signal is not further declined by its shorter length in multi-turn interactions; its contribution to the batch gradient is simply proportional to its token share (20–30%), instead of becoming negligible.
> >
> > 3. **Empirically, DFPO effectively learns the draft.** DFPO consistently outperforms M-GRPO [3] and DAPO under the same RL training budget (Table 2 in the paper). These gains are obtained with the current draft lengths and training scheme, indicating that the draft supervision is sufficiently strong in practice and materially contributes to the overall improvement.
> >
> > References:
> >
> > [1] Understanding R1-Zero-Like Training: A Critical Perspective, COLM 2025.
> >
> > [2] DAPO: An Open-Source LLM Reinforcement Learning System at Scale, NeurIPS 2025.
> >
> > [3] WebAgent-R1: Training Web Agents via End-to-End Multi-Turn Reinforcement Learning, EMNLP 2025 Main.

---

> > > ### Author Response · Authors · 2025-11-21
> > > **Response to Reviewer YEmS (Appendix)**
> > >
> > > **checkpoint: Qwen 2.5-7B-Instruct; seed: 42; dataset: AirQA-Real/SciDQA**
> > >
> > > | Method | Avg. | I-Avg. | Valid Answers | Interaction Turns |
> > > | --- | --- | --- | --- | --- |
> > > | **DFPO w/DTFT** | 25.3/48.3 | 21.0/37.4 | 85.4/80.7 | 6.20/8.01 |
> > > | **SFT** | 23.9/45.6 | 17.8/33.2 | 71.6/65.3 | 8.94/9.39 |
> > >
> > > **checkpoint: Qwen 2.5-3B-Instruct; seed: 42; dataset: AirQA-Real/SciDQA**
> > >
> > > | Method | Avg. | I-Avg. | Valid Answers | Interaction Turns |
> > > | --- | --- | --- | --- | --- |
> > > | **DFPO w/DTFT** | 23.7/43.5 | 20.0/36.1 | 88.1/84.5 | 5.75/6.23 |
> > > | **SFT** | 22.2/43.3 | 16.7/28.4 | 70.3/56.0 | 8.74/11.4 |
> > >
> > > **checkpoint: Qwen 2.5-7B-Instruct; seed: 0; dataset: AirQA-Real/SciDQA**
> > >
> > > | Method | Avg. | I-Avg. | Valid Answers | Interaction Turns |
> > > | --- | --- | --- | --- | --- |
> > > | **DFPO w/DTFT** | 25.0/48.6 | 20.8/37.6 | 84.9/81.2 | 6.16/8.03 |
> > > | **SFT** | 23.2/44.7 | 17.1/32.2 | 69.3/62.7 | 9.13/9.62 |
> > >
> > > **checkpoint: Qwen 2.5-3B-Instruct; seed: 0; dataset: AirQA-Real/SciDQA**
> > >
> > > | Method | Avg. | I-Avg. | Valid Answers | Interaction Turns |
> > > | --- | --- | --- | --- | --- |
> > > | **DFPO w/DTFT** | 23.9/44.0 | 20.1/36.5 | 86.7/83.9 | 5.86/6.40 |
> > > | **SFT** | 22.1/42.7 | 16.6/28.0 | 73.4/56.6 | 8.61/11.4 |
> > >
> > > **checkpoint: Qwen 2.5-7B-Instruct; seed: 1; dataset: AirQA-Real/SciDQA**
> > >
> > > | Method | Avg. | I-Avg. | Valid Answers | Interaction Turns |
> > > | --- | --- | --- | --- | --- |
> > > | **DFPO w/DTFT** | 25.7/47.8 | 21.2/37.0 | 86.0/79.9 | 6.39/8.02 |
> > > | **SFT** | 23.4/45.0 | 17.2/32.6 | 69.1/64.1 | 9.19/9.76 |
> > >
> > > **checkpoint: Qwen 2.5-3B-Instruct; seed: 1; dataset: AirQA-Real/SciDQA**
> > >
> > > | Method | Avg. | I-Avg. | Valid Answers | Interaction Turns |
> > > | --- | --- | --- | --- | --- |
> > > | **DFPO w/DTFT** | 24.0/43.8 | 19.9/35.9 | 85.9/84.4 | 6.07/6.56 |
> > > | **SFT** | 21.8/42.8 | 16.4/28.0 | 71.0/57.8 | 8.92/11.2 |
> > >
> > > **checkpoint: Qwen 2.5-7B-Instruct; seed: 2; dataset: AirQA-Real/SciDQA**
> > >
> > > | Method | Avg. | I-Avg. | Valid Answers | Interaction Turns |
> > > | --- | --- | --- | --- | --- |
> > > | **DFPO w/DTFT** | 24.8/48.0 | 20.6/37.3 | 83.1/81.0 | 6.20/7.93 |
> > > | **SFT** | 22.7/44.5 | 16.8/32.3 | 67.9/62.9 | 9.04/9.53 |
> > >
> > > **checkpoint: Qwen 2.5-3B-Instruct; seed: 2; dataset: AirQA-Real/SciDQA**
> > >
> > > | Method | Avg. | I-Avg. | Valid Answers | Interaction Turns |
> > > | --- | --- | --- | --- | --- |
> > > | **DFPO w/DTFT** | 23.5/43.8 | 19.9/36.2 | 87.0/85.1 | 6.13/6.33 |
> > > | **SFT** | 21.5/42.7 | 16.3/27.9 | 71.9/57.3 | 8.50/11.5 |
> > >
> > > **checkpoint: Qwen 2.5-7B-Instruct; seed: 3; dataset: AirQA-Real/SciDQA**
> > >
> > > | Method | Avg. | I-Avg. | Valid Answers | Interaction Turns |
> > > | --- | --- | --- | --- | --- |
> > > | **DFPO w/DTFT** | 25.7/49.0 | 21.3/38.0 | 85.8/81.9 | 6.26/7.97 |
> > > | **SFT** | 23.7/46.2 | 17.3/33.0 | 71.0/65.2 | 9.34/9.79 |
> > >
> > > **checkpoint: Qwen 2.5-3B-Instruct; seed: 3; dataset: AirQA-Real/SciDQA**
> > >
> > > | Method | Avg. | I-Avg. | Valid Answers | Interaction Turns |
> > > | --- | --- | --- | --- | --- |
> > > | **DFPO w/DTFT** | 23.7/43.7 | 19.9/36.2 | 86.9/83.9 | 5.89/6.28 |
> > > | **SFT** | 20.9/42.8 | 15.8/28.0 | 73.0/58.1 | 8.64/11.3 |

---

> > > > ### Comment · Reviewer_YEmS · 2025-11-26
> > > >
> > > > Thank you for the detailed response. The authors' reply and experiments have addressed most of my concerns. However, I still believe that Theorem 1 is unnecessary and remains a product of algebraic rearrangement. It is insufficient to warrant the formulation as Theorem 1. I have also carefully reviewed the questions from other reviewers and the authors' responses. Ultimately, I have decided to raise my rating to 6 and lower my confidence to 2. I am raising the rating because I believe the authors' idea and experiments are both strong and thorough. I am lowering my confidence because this type of presentational packaging in the writing undermines the paper's credibility and readability.

---

> > > > > ### Author Response · Authors · 2025-11-27
> > > > > **Restating Theorem 1 as Proposition 1**
> > > > >
> > > > > We sincerely thank you again for your positive assessment of PaperCompass!
> > > > >
> > > > > As we have emphasized, the goal of our mathematical derivations is not to over-package an engineering implementation, but to provide an analysis of DFPO that helps readers better understand its behavior and our contributions.  We fully understand your concerns about the presentation and are grateful for your constructive suggestions regarding the paper's credibility and readability.  In the revised manuscript that we will upload before the rebuttal deadline, we will accordingly **restate Theorem 1 as Proposition 1** and clarify its explanatory role.

---

### Official Review · Reviewer_pyt2 · 2025-11-03

**Soundness:** 3
**Presentation:** 3
**Contribution:** 2
**Rating:** 6
**Confidence:** 3

**Summary:**

In this paper, the authors propose a multi-turn RL framework called PaperCompass for training agents for scientific paper querying. Authors take inspiration from cognitive science to bridge the knowing-doing gap by constructing high-level plans, and then getting into more fine-grained actions. They propose a novel DFPO method to train for this method, imposing a hierarchical decision-making process: A high-level
draft leverages the LLM’s instruction-following capabilities to guide the low-level follow execution, thereby preventing deviations into greedy action sequences.

**Strengths:**

Strengths:
1. Novel hierarchical decision-making approach
2. Theoretical results add more clarity and justification to the proposed approach.

**Weaknesses:**

Weaknesses:
1. The authors should explain the knowing-doing gap concept more at the introduction of the paper.
2. Related work is not present in the main paper.
3. The details of the synthetic data generation are also deferred to the Appendix. I think the authors should add one paragraph to show how synthetic data is being generated.
4. Only the Qwen family is considered for finetuning the case.

**Questions:**

Questions:
1. Why is only one LLM agent used? What happens if two LLM agents are used to generate the draft and to execute the plan, respectively?
2. On line 408 the authors say that the result is interpreted as the narrowing of knowing doing gap, but the rationale behind this interpretation is not clear. Any metrics to show that?

---

> ### Author Response · Authors · 2025-11-21
> **Response to Reviewer pyt2 (1/2)**
>
> We thank the reviewer for the constructive feedback and address the comments below. We have highlighted the revised and newly added clarifications in **yellow** in the revised manuscript.
>
> ## **Weakness 1: Further explaining ‘knowing-doing’ gap in Introduction**
>
> We have strengthened the discussion of the knowing–doing gap in the Introduction by:
>
> 1. Providing a concrete explanation of how the phenomenon manifests specifically in Paper-QA tasks (e.g., repeated retrieval loops despite correct reasoning).
> 2. Making the motivation clearer by explicitly connecting knowing–doing gap to the need for structurally separating reasoning (“knowing”) from execution (“doing”) in our framework.
>
> These clarifications have been incorporated into the revised Introduction.
>
> ## **Weakness 2: Related work is not present in the main paper**
>
> In the initial submission, we placed the Related Work section in the Appendix due to the 9-page limit. During the rebuttal phase, the page limit is extended to 10 pages, which allowed us to integrate the full Related Work section back into the main text. The revised version now includes a dedicated Related Work section in the main paper (Section 5).
>
> ## **Weakness 3: Add one paragraph to show how synthetic data is being generated**
>
> We agree that the main paper should provide a concise overview of how the synthetic expert trajectories are generated. In the revised version, we added a short, high-level description of the data synthesis pipeline to the main paper (Section 3. 2), summarizing the roles of the Explorer, Actor, and Tracker and how trajectories and drafts are constructed.
>
> ## **Weakness 4: Only Qwen family is considered for finetuning the case**
>
> Thank you for pointing out this limitation. To address your concern, we fine-tuned and conducted **a new set of experiments on Llama3.1-8B-Instruct**, a model fully disjoint from Qwen and never used in our development pipeline. The results show that **PaperCompass generalizes beyond the Qwen family**, consistently improving interaction efficiency while preserving (or slightly improving) accuracy.
>
> **Main Results on AirQA-Real**
>
> | **Method** | text | table | image | form. | meta. | **Avg.** | **I-Avg.** |
> | --- | --- | --- | --- | --- | --- | --- | --- |
> | **Llama3.1-8B-Instruct** | 7.2 | 0.0 | 0.0 | 5.6 | 4.5 | 6.7 | / |
> | **Llama3.1-8B-Instruct w/ SFT** | 23.8 | 4.0 | 12.5 | 13.9 | 9.1 | 21.9 | 16.6 |
> | **Llama3.1-8B-Instruct w/ DTFT** | 23.8 | 3.0 | 15.0 | 8.3 | 13.6 | 22.2 | 17.1 |
> | **PaperCompass-8B** | 27.9 | 4.0 | 12.5 | 13.9 | 13.6 | 24.8 | 20.0 |
>
> **Efficiency Statistics on AirQA-Real**
>
> | **Method** | Correct Rate | Valid Answers | Interaction Turns |
> | --- | --- | --- | --- |
> | **Llama3.1-8B-Instruct w/ SFT** | 21.9 | 75.6 | 8.45 |
> | **Llama3.1-8B-Instruct w/ DTFT** | 22.2 | 76.5 | 8.09 |
> | **PaperCompass-8B** | 24.8 | 81.4 | 6.97 |
>
> **Main Results on SciDQA**
>
> | **Method** | text | table | image | form. | **Avg.** | **I-Avg.** |
> | --- | --- | --- | --- | --- | --- | --- |
> | **Llama3.1-8B-Instruct** | 45.9 | 45.1 | 45.6 | 43.8 | 45.8 | / |
> | **Llama3.1-8B-Instruct w/ SFT** | 45.2 | 46.2 | 46.1 | 41.7 | 46.1 | 29.9 |
> | **Llama3.1-8B-Instruct w/ DTFT** | 47.6 | 47.9 | 47.4 | 46.2 | 47.6 | 33.0 |
> | **PaperCompass-8B** | 47.3 | 48.1 | 47.6 | 46.0 | 47.4 | 36.0 |
>
> **Efficiency Statistics on SciDQA**
>
> | **Method** | Correct Rate | Valid Answers | Interaction Turns |
> | --- | --- | --- | --- |
> | **Llama3.1-8B-Instruct w/ SFT** | 46.1 | 55.1 | 11.61 |
> | **Llama3.1-8B-Instruct w/ DTFT** | 47.6 | 61.7 | 10.36 |
> | **PaperCompass-8B** | 47.4 | 74.2 | 8.45 |
>
> Across both datasets, PaperCompass-8B yields **substantial efficiency gains** (e.g., I-Avg: 16.6 → 20.0; Interaction Turns: 8.45 → 6.97 on AirQA-Real, and similar gains on SciDQA), while maintaining accuracy at or above SFT/DTFT baselines. These trends closely mirror those observed on the Qwen family. These results suggest that our method is **not tied to the Qwen family**: PaperCompass remains effective when transferred to Llama3.1-8B-Instruct, despite differences in architecture, tokenizer, and alignment pipeline.

---

> > ### Author Response · Authors · 2025-11-21
> > **Response to Reviewer pyt2 (2/2)**
> >
> > ## **Question 1: Why only one LLM agent used? What happens if two LLM agents are used?**
> >
> > 1. **Learning simultaneously shapes cognition and action within a single agent.** Where learning has an impact lies in both cognition and action, that is, the agent’s ability to recognize and to act [1]. Learning and optimization modify internal structures or components within the agent, ultimately influencing its decision-making. Our goal is to enable the **same agent** to optimize both cognition and action simultaneously, which is precisely what we emphasize in the paper as addressing the “knowing–doing gap.”
> > 2. **Why a separated planner–executor architecture cannot achieve our goal?** Some works [2,3,4,5] adopt a separated structure of planner and executor, often using a more advanced, larger model as the planner. However, this hierarchy is **static**, meaning that the plan and the execution must be optimized separately. Such static separation prevents end-to-end learning of the entire decision-making process and makes it impossible for the system to **jointly** optimize cognition and action in a unified manner.
> > 3. **Why hierarchical RL (e.g., ArCHer) still does not satisfy our objective?** Of course, one may consider hierarchical reinforcement learning algorithms (such as ArCHer [6]) that optimize the top-level agent and the low-level agent separately. However, these approaches still optimize **different components in isolation**: the high-level agent optimizes cognition, while the low-level agent optimizes action. This structure still cannot achieve our goal of jointly improving cognition and action **within a single agent**, nor can it address the knowing–doing gap that our framework targets.
> >
> > References:
> >
> > [1] Advances and Challenges in Foundation Agents: From Brain-Inspired Intelligence to Evolutionary, Collaborative, and Safe Systems, arXiv: 2504.01990.
> >
> > [2] HiRA: A Hierarchical Reasoning Framework for Decoupled Planning and Execution in Deep Search, AAAI 2026.
> >
> > [3] Divide and Conquer: Grounding LLMs as Efficient Decision-Making Agents via Offline Hierarchical Reinforcement Learning, ICML 2025.
> >
> > [4] Learning to plan before answering: Self-teaching llms to learn abstract plans for problem solving, ICLR 2025.
> >
> > [5] Distilling LLM Agent into Small Models with Retrieval and Code Tools, NeurIPS 2025.
> >
> > [6] ArCHer: Training Language Model Agents via Hierarchical Multi-Turn RL, ICML 2024.
> >
> > ## **Question 2: Any metric to show the narrowing of knowing-doing gap?**
> >
> > Empirically, in Appendix F we provide case studies comparing PaperCompass-7B with Qwen2.5-14B-Instruct. PaperCompass-7B consistently follows the draft plan and reaches the correct answer, whereas Qwen2.5-14B-Instruct, without draft guidance, exhibits typical failure modes such as prematurely terminating the search or retrieving the wrong content.
> >
> > To measure the **knowing–doing gap**, we used a UCB-based diagnostic bandit task [1]. In each episode, the agent is shown 3 arms, each associated with a hidden reward distribution. The agent must (i) compute the value scores provided in the prompt (“knowing’’) and (ii) select an arm (“doing’’). The optimal arm is uniquely determined from the scores, so a mismatch between reasoning and action directly reflects a knowing–doing gap. Formally, we defined the knowing-doing gap as **P(action is non-optimal | knowing=True)**. We evaluated the model over 64 independent environments, each with 50 decision steps, and reported how often the model correctly identifies the optimal arm but fails to choose it. We have also **included the experimental results in the revised version**. You can find the instruction in Appendix A.4.5.
> >
> > **Qwen2.5-7B-Instruct**
> >
> > |  | **Action: Optimal** | **Action: Greedy** | **Action: Other** |
> > | --- | --- | --- | --- |
> > | **Knowing: True** | 5.7 | 1.9 | 6.1 |
> > | **Knowing: False** | 2.9 | 63.9 | 19.6 |
> >
> > **PaperCompass-7B**
> >
> > |  | **Action: Optimal** | **Action: Greedy** | **Action: Other** |
> > | --- | --- | --- | --- |
> > | **Knowing: True** | 8.3 | 4.6 | 3.6 |
> > | **Knowing: False** | 2.4 | 53.9 | 27.2 |
> >
> > As shown in the tables above, PaperCompass-7B not only achieves a higher rate of correct reasoning (16.5% vs. 13.7%), but also conditioned on the steps where the model’s reasoning is correct (Knowing=True), it also converts that reasoning into correct actions more reliably: **8.3/16.5 = 50.3%** for PaperCompass-7B vs. **5.7/13.7 = 41.6%** for Qwen2.5-7B-Instruct. This directly indicates a reduction in the knowing–doing gap.
> >
> > **References**:
> >
> > [1] LLMs are Greedy Agents: Effects of RL Fine-tuning on Decision-Making Abilities, ICML 2025 Workshop EXAIT.

---

> > > ### Author Response · Authors · 2025-11-28
> > > **We are looking forward to further discussing with you**
> > >
> > > Dear Reviewer pyt2,
> > >
> > > I hope this message finds you well. As the discussion period is nearing its end with **less than one week remaining**, I wanted to ensure we have addressed all your concerns satisfactorily. If there are any additional points or feedback you would like us to consider, please let us know. Your insights are invaluable to us, and we are eager to address any remaining issues to improve our work.
> > >
> > > Thank you for your time and effort in reviewing our paper!
> > >
> > > Best regards,
> > >
> > > The authors of Submission 3227

---

### Author Response · Authors · 2025-11-24
**General Response**

We sincerely thank all reviewers for their thoughtful and constructive feedback. We are encouraged by their recognition of the key ideas in our work, including the Draft-and-Follow hierarchical decision-making framework, the DFPO algorithm for jointly optimizing draft planning and solution execution, and the substantial interaction-efficiency gains demonstrated on Paper-QA tasks. For clarity and simplicity, we refer to Reviewers **pyt2**, **YEmS**, **cwPa**, and **UDhF** as R1, R2, R3, and R4, respectively, in the following response. All revisions and newly added content are highlighted in yellow.

---

### **Core Contributions of Our Work**

- **Draft-and-Follow Agent Framework.** We propose Draft-and-Follow architecture for multi-turn paper QA, where a high-level draft provides coarse planning and the subsequent solution follows ReAct-style grounded execution. This structure targets the knowing–doing gap in small LLM agents by explicitly separating “knowing” (planning) from “doing” (tool-grounded action) while enabling joint optimization.

- **DFPO: Joint Optimization of Draft and Solution.** We introduce DFPO (Draft-and-Follow Policy Optimization), an RL fine-tuning algorithm that decomposes gradients for draft vs. solution under unified normalization, yielding an implicit draft-bias term that encourages plans to be both correct and action-aligned. DFPO improves long-horizon consistency and reduces repetitive retrieval loops.

- **Strong Empirical Efficiency Gains with Preserved Accuracy.** Across AirQA-Real and SciDQA, DFPO consistently improves interaction efficiency while maintaining or slightly improving accuracy, showing that small agents can achieve high research utility with fewer tool calls.

---

### **Updates of Experimental Results During Rebuttal**

- **Cross-family generalization (R1).** We added new fine-tuning and evaluation on Llama3.1-8B-Instruct. PaperCompass-8B reproduces the same pattern of efficiency gains with comparable accuracy on both AirQA-Real and SciDQA, demonstrating that our method is not tied to the Qwen family.

- **Cross-domain QA generalization (R4).** We evaluated Qwen2.5-7B-Instruct w/ SFT and PaperCompass-7B on HotpotQA and MuSiQue using the Search-R1 interaction protocol, without any in-domain training on these datasets. PaperCompass-7B generalizes better than the SFT baseline under identical tools/settings.

- **Statistical significance validation (R2).** We re-ran all main experiments with 5 random seeds for both 3B and 7B models on AirQA-Real and SciDQA. We report mean ± std and paired t-tests, showing DFPO consistently outperforms SFT with statistically significant improvements across all core metrics.

- **Draft length and training-signal analysis (R2).** We added detailed draft/solution token statistics. Drafts occupy ~20–30% of full trajectories, and unified per-token normalization in DFPO ensures draft gradients are not drowned out. Empirical gains confirm sufficient learning signal.

- **Human-centered utility evaluation (R3).** We added an expert trajectory study on 20 sampled questions. Expert rates PaperCompass-7B higher in retrieval quality, tool-use soundness, and overall usefulness, complementing automatic metrics.

- **Robustness to unreliable tools (R4).** We introduced three types of tool noise (call failure, top-k corruption, structural noise), retrained PaperCompass-7B-Noise, and showed it degrades less than the SFT baseline under the same noisy test environment.

---

### **Updates of In-depth Discussions During Rebuttal**

- **Clearer motivation and definition of knowing–doing gap (R1).** We strengthened the Introduction with a concrete Paper-QA-specific explanation of the gap (e.g., repeated retrieval loops despite correct reasoning) and clarified why structural draft/solution separation is required.

- **Refined theoretical framing and narrative (R2).** We revised wording to avoid the misleading impression that DFPO introduces a new explicit “regularizer.” We emphasize Theorem 1’s role as explanatory, highlighting that the extra term is an implicit bias from unified normalization and prefix-draft structure. For negative sample masking, we clarified the non-circular proof flow: Theorem 2 (no masking) motivates masking as a practical stabilization choice, followed by Proposition 1/Lemma 3 analyzing effects conditional on masking.

- **Training cost transparency (R4).** We added a precise compute breakdown (NPU-hours) comparing DTFT+DFPO vs. SFT+GRPO, showing ~1.7× overhead under matched steps/hardware, and discussed amortization via downstream efficiency savings.

---

We believe these additions and revisions comprehensively address the reviewers’ concerns and substantially improve the clarity, rigor, and generality of the paper. We are grateful again for the valuable feedback and look forward to the reviewers’ favorable consideration.

---

### Author Response · Authors · 2025-11-30
**Additional note to AC**

We would like to add a brief clarification following the review reversion.

**Reviewer YEmS** initially gave an overall score of 4. In the later official comment posted on Nov 26, the reviewer explicitly states that most concerns have been addressed by our rebuttal and the new experiments, and therefore decides to **raise the rating to 6 while lowering the confidence to 2**. Consequently, after the discussion phase, the effective consensus across the four reviewers is  **6 / 6 / 6 / 6**.

All technical clarifications and additional results (cross-family and cross-task generalization, statistical significance analysis, human-centred evaluation, robustness under noisy tool environments, and the training-cost breakdown) have been summarized in our previously posted *General Response*. We hope this information will be helpful for your meta-review, and we are grateful for your time and consideration.

---

### Note · Program_Chairs · 2026-01-17
**Submission Desk Rejected by Program Chairs**

The following references in this submission do not refer to real documents and/or have major errors in bibliographic information:

 Jiawei Wu, William W. Cohen, and Pradeep Ravikumar. Mineru: A general-purpose and extensible platform for document-level element extraction. In Findings of the Association for Computational Linguistics: EMNLP 2023, pp. 9548-9562, Singapore, dec 2023.